# Suppressing peatland methane production by electron snorkeling through pyrogenic carbon in controlled laboratory incubations

Tianran Sun[1,2], Juan J. L. Guzman[3], James D. Seward[4], Akio Enders[1], Joseph B. Yavitt[5], Johannes Lehmann [1,6] & Largus T. Angenent [2,3,6✉]

Northern peatlands are experiencing more frequent and severe fire events as a result of changing climate conditions. Recent studies show that such a fire-regime change imposes a direct climate-warming impact by emitting large amounts of carbon into the atmosphere. However, the fires also convert parts of the burnt biomass into pyrogenic carbon. Here, we show a potential climate-cooling impact induced by fire-derived pyrogenic carbon in laboratory incubations. We found that the accumulation of pyrogenic carbon reduced post-fire methane production from warm (32 °C) incubated peatland soils by 13–24%. The redox-cycling, capacitive, and conductive electron transfer mechanisms in pyrogenic carbon functioned as an electron snorkel, which facilitated extracellular electron transfer and stimulated soil alternative microbial respiration to suppress methane production. Our results highlight an important, but overlooked, function of pyrogenic carbon in neutralizing forest fire emissions and call for its consideration in the global carbon budget estimation.

[1] Soil and Crop Sciences, School of Integrative Plant Science, College of Agriculture and Life Sciences, Cornell University, Ithaca, NY, USA. [2] Center for Applied Geosciences, University of Tübingen, Tübingen, Germany. [3] Department of Biological and Environmental Engineering, College of Agriculture and Life Sciences, Cornell University, Ithaca, NY, USA. [4] Vale Living with Lakes Centre and the Department of Biology, Laurentian University, Sudbury, ON, Canada. [5] Department of Natural Resources, Cornell University, Ithaca, NY, USA. [6] Atkinson Center for a Sustainable Future, Cornell University, Ithaca, NY, USA. ✉email: l.angenent@uni-tuebingen.de

Multiple lines of evidence show that forest fires become increasingly frequent in northern peatlands (i.e., latitude 40°–70°N) due to their vulnerability to a warming and drying climate[1,2]. Increasing fires threaten to shift the soil carbon regime from a sink to a source by releasing large amounts of carbon into the atmosphere[3], but they also convert parts of the burnt biomass into pyrogenic carbon through the incomplete combustion of biomass[4]. Due to its carbon enrichment and environmental persistence, pyrogenic carbon has shown the ability to buffer carbon emission during the fire[5] and store carbon in peat soils from centuries to millennia after the fire[6–8]. In this context, current studies are striving to quantify the production of pyrogenic carbon caused by fires[9] and fill up the storage gap in the global carbon budget[8]. However, thus far, it is still unclear how this long-term stored pyrogenic carbon interacts with the indigenous methane ($CH_4$) production in peat soils.

Northern peatlands have historically been a sink for atmospheric carbon dioxide ($CO_2$) but a source of atmospheric $CH_4$[10]. Even though methanogens in peat soils are phylogenetically diverse, they produce $CH_4$ mainly through hydrogenotrophic ($\Delta G^0 = -33$ kJ mol$^{-1}$ hydrogen) and aceticlastic ($\Delta G^0 = -36$ kJ mol$^{-1}$ acetate) methanogenesis[11]. The prevalence of either methanogenesis pathways depends on the nutrient level and feeding type (i.e., whether it is rain-fed or stream-fed) of specific peat soil, but they all compete with anaerobic respiration that utilizes alternative (to oxygen) terminal electron acceptors (hereafter alternative respiration) for substrate[11]. Alternative respiration has been widely studied and demonstrated its favorability ($\Delta G^0 = -40$ to $-234$ kJ mol$^{-1}$ hydrogen and $-69$ to $-841$ kJ mol$^{-1}$ acetate) in substrate competition with methanogenesis, and thus suppressing $CH_4$ production[12,13]. Alternative terminal electron acceptors that have been reported in peat soils include manganese oxide, nitrate, iron minerals, sulfate, and dissolved and particulate organic matter in the order of high to low reduction potential (averagely from 0.8 to $-0.2$ V[14]). Specialized anaerobic microbes, such as Geobacter and Shewanella species, can perform alternative respiration by transferring electrons out of the cell (i.e., extracellular electron transfer) to solid-phase alternative terminal electron acceptors and gain energy for their growth[15]. These microbes can also use pyrogenic carbon as an intermediate electron acceptor and conduit through redox-active quinone and hydroquinone functional groups and electrically conductive carbon matrices in pyrogenic carbon[16,17].

Electron accepting and donating cycles of the pyrogenic carbon quinone and hydroquinone functional groups have been shown to mediate biogeochemical redox reactions in facilitating microbial mineral reduction[18] and adjusting nitrification and denitrification cycling[19]. The pyrogenic carbon matrices possess a wide 1.2 V potential range to directly transfer electrons to a variety of terminal electron acceptors, including manganese oxide, iron minerals, and protons[17]. These demonstrated abilities to take and pass electrons from and to different environmental substances led us to hypothesize that: (1) pyrogenic carbon can function as a microbial electron-snorkel[20] in anaerobic peat soil, which stimulates alternative respiration by facilitating extracellular electron transfer to reach a larger electron-accepting pool; and (2) the enhanced alternative respiration can outcompete methanogenesis and potentially reduces peatland net $CH_4$ emissions. To verify these hypotheses, we conducted 6 large-group microcosm and bioelectrochemical peat soil as well as pure-culture incubations, which in total contained 144 individual incubations to probe the functions of pyrogenic carbon in electron snorkeling and $CH_4$ suppression. This systematic study helped to better understand the post-fire interactions between pyrogenic carbon and methanogens as well as the underlying mechanisms controlling $CH_4$ production, which might improve

the estimates, and more importantly, the prediction of peatland gas emissions for future climate change scenarios.

## Results

**Electron snorkeling suppressed methane production in peat soil.** We tested the electron snorkeling of pyrogenic carbon and its effect on methanogenesis in artificial microcosm and bioelectrochemical peat-soil incubations (Supplementary Method 1–4). The soil was sampled from an ombrotrophic (i.e., rain fed) peat site that is known to produce $CH_4$, with a field production rate of 3.5 Mg ha$^{-1}$ yr$^{-1}$ ref. [21], projecting to 119 Mg $CO_{2equivalent}$ ha$^{-1}$ yr$^{-1}$ (assuming a global warming potential of methane over 100 years). Hydrogenotrophic methanogens (Methanoregula and Methanocellales, with an average abundance of 1.2% of total microbial population for each) were found to dominate the peat soil $CH_4$ production. Geobacter spp. (with an average abundance of 1% of total microbial population) was determined as the major microbial member that performed alternative respiration (Supplementary Fig. 1). We performed microcosm and bioelectrochemical peat-soil incubations at 32 °C in an incubation room to maintain an optimum temperature for methanogens' metabolism in the studied peat soil[22] and access the upper potential of the amount of $CH_4$ that can be suppressed by the accumulation of pyrogenic carbon.

Pyrolyzed biomass (considering various combustion temperatures ranging from 400 to 800 °C) was added to the peat soil to recreate the natural accumulation of pyrogenic carbon after vegetation fires. To distinguish pyrogenic carbon from native soil carbon and investigate the effect of biodegradation of pyrogenic carbon on gas production, the original biomass was isotopically labeled with $^{13}C$, which resulted in a $\delta^{13}C$ of pyrogenic carbon at $774 \pm 2.3‰$ (Supplementary Method 5). We added pyrogenic carbon at concentrations (0, 0.03, 3, and 10 mg pyrogenic carbon g$^{-1}$ peat soil) that lie in the lower range of its naturally occurring concentration in peat soils[8] to conservatively assess the effect of electron snorkeling on $CH_4$ suppression. Even though the added concentration varied among incubations, we normalized the number of snorkeled electrons to a unit mass of either pyrogenic carbon or soil carbon to focus on the electron snorkeling efficiency instead of the effect of different addition rates. We found that three electron transfer mechanisms contributed to electron snorkeling of pyrogenic carbon and caused significant decreases in $CH_4$ production: (1) redox-cycling electron transfer by the quinone and hydroquinone functional groups; (2) capacitive electron transfer through the carbon matrices; and (3) conductive electron transfer through the carbon matrices. While redox-cycling electron transfer was the kinetically preferred mechanism for the electron snorkeling of pyrogenic carbon that was produced at low-temperature range, capacitive and conductive electron transfer mechanisms dominated the electron snorkeling of pyrogenic carbon that was produced at high-temperature range (Fig. 1 and Supplementary Fig. 2).

We investigated the effect of redox-cycling electron transfer on $CH_4$ production in the microcosm peat-soil incubations (Supplementary Fig. 3a). Results showed that addition of pyrogenic carbon (produced at 400 and 500 °C pyrolysis temperatures) caused an instant drop in $CH_4$ production, resulting in a 13% ($N = 3$, $P = 0.03$) lower total production ($41 \pm 2.1$ μmol g$^{-1}$ soil carbon, orange and green lines in Fig. 1a) than the pyrogenic carbon-free control treatment ($47 \pm 2.5$ μmol g$^{-1}$ soil carbon, black line in Fig. 1a). Along with the decrease of $CH_4$ production, we observed an increased $CO_2$ accumulation (Supplementary Fig. 2), which suggested that the fate of $CH_4$ was determined by microbial processes rather than adsorption. In accordance with the $CH_4$ suppression, we observed an increased net accumulation

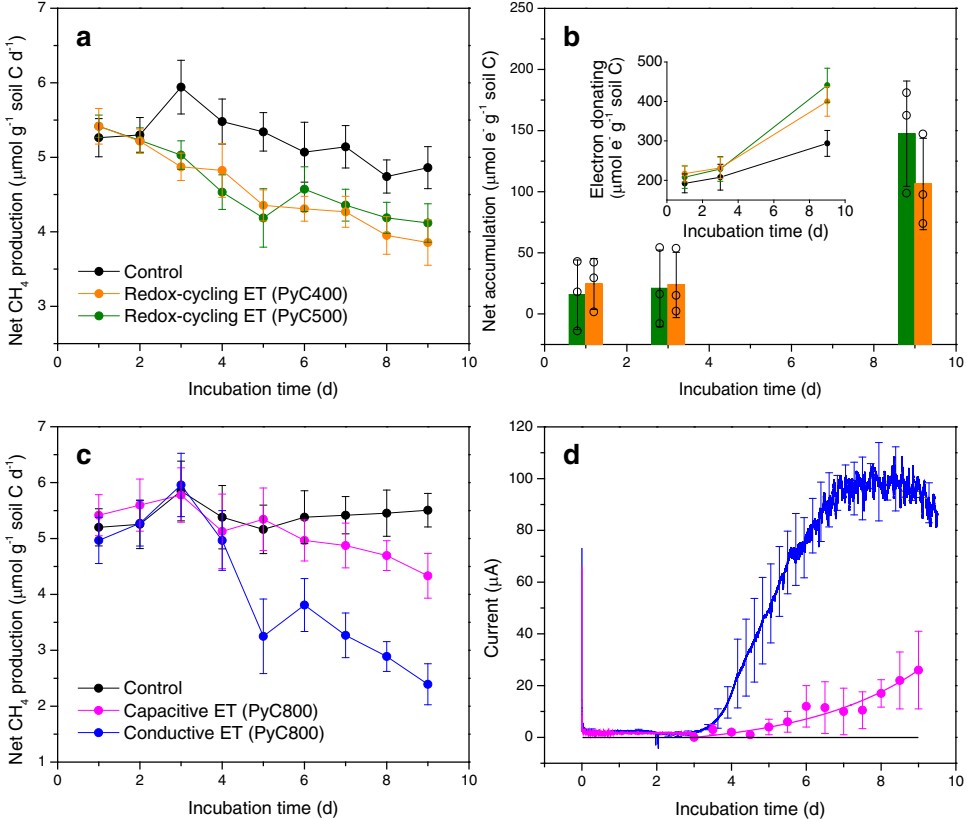

**Fig. 1 Electron snorkeling influenced CH$_4$ production in the peat-soil incubations (at 32 °C and dark). a** Daily CH$_4$ production rate in the microcosm peat-soil incubations that were associated with the redox-cycling electron transfer (ET) of the pyrogenic carbon functional groups. Figure legends in chart **a** also applies to chart **b**. **b** Net electron accumulation caused by the redox-cycling ET in peat soil organic matter at the incubation day 1, 3, and 9. Open circles indicate the corresponding data points of the bar chart. The net electron accumulation was calculated based on the electron-donating difference (inset chart) between the pyrogenic carbon-amended peat soil and the control peat soil. **c** Daily CH$_4$ production rate in the bioelectrochemical peat-soil incubations that were associated with the capacitive and conductive ET through the pyrogenic carbon matrices. Figure legends in chart **c** also applies to chart **d**. **d** Current profiles generated from the capacitive and conductive ET through the pyrogenic carbon matrices. The intermittent current signals in the capacitive ET showed the highest current point of each discharging period at a 0.1 s$^{-1}$ recording frequency. Full discharging current profile could be found at the chronoamperograms in Supplementary Fig. 6. Day 0 in the x axes in **a–d** indicate the time of adding pyrogenic carbon into the peat soil. Error bars are s. d. of triplicate measurements. For significant differences, see text. Data are presented as mean values ± s.d.

of electrons (147 ± 43 µmol e$^-$ g$^{-1}$ soil carbon) in the pyrogenic carbon-amended peat soil in comparison with the pyrogenic carbon-free control treatment (Fig. 1b and Supplementary Fig. 4). We further identified that soil organic matter dominated the electron-accepting capacity of the peat soil and was responsible for accumulating electrons (Supplementary Method 1).

Appearance of electron accumulation suggested that the alternative respiration had been stimulated due to the presence of pyrogenic carbon because more electrons were released into the peat soil. Redox-active pyrogenic carbon stimulated alternative respiration by facilitating extracellular electron transfer through the electron-accepting capability of its quinone groups. More importantly, we found that the number of accumulated electrons (74 ± 22 µmol e$^-$) in the peat-soil organic matter was eightfold ($N = 3$, $P < 0.01$) higher than the electron-accepting capacity (9.1 ± 0.7 µmol e$^-$, Supplementary Method 6) of pyrogenic carbon quinone groups. This excessive electron-accepting suggested that after reaching its maximum capacity, pyrogenic carbon could be regenerated by donating (in the form of hydroquinone) the accepted electrons to soil organic matter. Such repeated redox-cycling electron transfer completed the electron snorkeling process of the pyrogen carbon quinone and hydroquinone groups and extended the utilizable electron-accepting capacity of the peat soil[23] to reduce CH$_4$ production.

In contrast to the quinone and hydroquinone groups, the pyrogenic carbon matrices (produced at 800 °C pyrolysis temperature) demonstrated a slow kinetics in transferring electrons to soil organic matter (Supplementary Fig. 5). However, even under this limited terminal electron-accepting condition, we were still able to observe a decreased CH$_4$ production after addition of pyrogenic carbon (red line in Fig. 1c). By using a bioelectrochemical circuit (Supplementary Fig. 3b) to periodically discharge pyrogenic carbon, we found that the electron accumulation (red line in Fig. 1d) that was corresponding to CH$_4$ suppression was occurring in the carbon matrices instead of soil organic matter. By integrating the discharging current (Supplementary Fig. 6) as a function of time, we quantified a total of 0.61 ± 0.31 mmol e$^-$ g$^{-1}$ pyrogenic carbon that was accumulated in the carbon matrices. We attributed this electron accumulation in the carbon matrices to stimulated alternative respiration, which donated electrons in a rate that was faster than electron accepting of soil organic matter. Such imbalanced electron fluxes that were transferred to and from the carbon matrices accomplished the capacitive electron transfer for electron snorkeling.

During capacitive electron transfer, the carbon matrices stimulated alternative respiration by storing extracellular electrons in the delocalized π-electron system in the crystalline graphitic structures. Existence of crystalline graphitic structures in the carbon

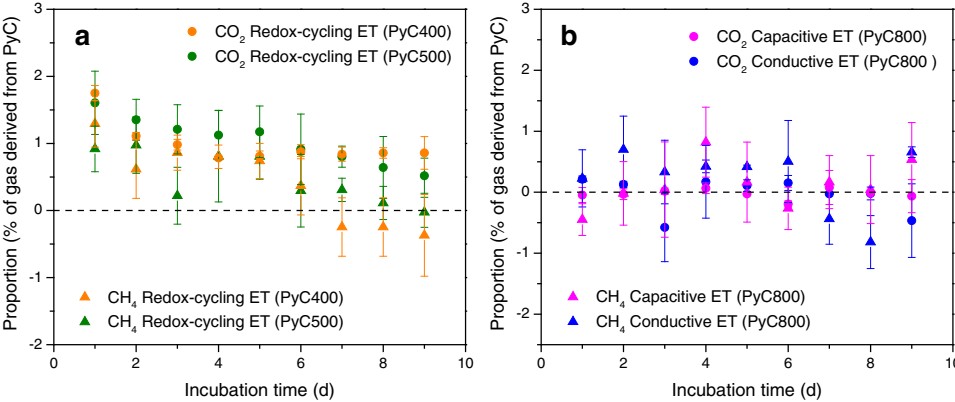

**Fig. 2 Proportions of $CO_2$ and $CH_4$ that were derived from the metabolism of pyrogenic carbon. a** Pyrogenic carbon-derived $CO_2$ and $CH_4$ production during the redox-cycling electron transfer (ET) of the pyrogenic carbon functional groups in the microcosm peat-soil incubations (at 32 °C and dark). Pyrogenic carbon was produced at 400 and 500 °C (PyC400 and PyC500) and the application rate was 3 mg pyrogenic carbon $g^{-1}$ soil. **b** Pyrogenic carbon-derived $CO_2$ and $CH_4$ production during the capacitive and conductive ET through the pyrogenic carbon matrices in the bioelectrochemical peat-soil incubations (at 32 °C and dark). Pyrogenic carbon was produced at 800 °C (PyC800) and the application rate was 10 mg pyrogenic carbon $g^{-1}$ soil. Dashed lines in **a** and **b** indicate the proportions generated from the pyrogenic carbon-free control treatments. Calculation of proportion can be found in Supplementary Method 5 and Supplementary Tables 1 and 2. Day 0 in the x-axes in **a** and **b** indicate the time of adding pyrogenic carbon into the peat soil. Error bars are s.d. of triplicate measurements. For significant differences, see text. Data are presented as mean values ± s.d.

matrices has been demonstrated by previous studies using X-ray and electron-based spectroscopic techniques[24,25]. Further abiotic cyclic voltammetric analysis confirmed that these structures possess a range of capacitance from 0.013 to 26 mF $cm^{-2}$ pyrogenic carbon with the increase of pyrolysis temperatures from 600 to 800 °C[17]. During alternative respiration, extracellular electrons were stored in the carbon matrices, which were surrounded and charge balanced by an ionic layer in the soil solution and constituted an electrical double-layer capacitance[26,27] for electron accumulation in the carbon matrices. It ought to be noted that the accumulated electrons comprised the overall electrons that were stored in the carbon matrices and in *Geobacter* spp. (most likely in a form of biofilm) respiring on the carbon matrices. The number of capacitively accumulated electrons (0.61 ± 0.31 mmol $e^-$ $g^{-1}$ pyrogenic carbon) was fourfold less ($N = 3$, $P = 0.01$) than the electrons (2.5 ± 0.7 mmol $e^-$ $g^{-1}$ pyrogenic carbon) that were accumulated in soil organic matter through the redox cycling of pyrogenic carbon during the same incubation period. Therefore, only at the late stage of the incubation did we observe a significant ($N = 3$, $P < 0.01$) decrease of $CH_4$ production (red line in Fig. 1c). Such an increase in suppression of $CH_4$ emission is expected to occur after an extended electron snorkeling period through capacitive electron transfer.

After the capacitance of the carbon matrices would be fully charged, a local electric field[27] across the pyrogenic carbon and peat soil interface could be built up and accelerate the electron accepting between the carbon matrices and soil organic matter. We found that accelerated electron accepting allowed a rapid and continuous transfer of extracellular electrons through the carbon matrices (tested with pyrogenic carbon produced at 800 °C pyrolysis temperature), which switched the electron snorkeling mechanism from the capacitive electron transfer to the conductive electron transfer (Fig. 1d). The carbon matrices are able to conductively transfer electrons due to the inherent electrical conductivity of $\pi$-electron system[17,28]. We implemented a bioelectrochemical circuit (Supplementary Fig. 3b) and successfully captured the current signal (blue line in Fig. 1d) while the conductive electron transfer was occurring. The current reached a peak of 70 μA $cm^{-2}$, which is in the range of electron transfer rate (20–250 μA $cm^{-2}$) of *Geobacter* species[29] but several orders of magnitude faster than the reported rate (0.1–1 μA $cm^{-2}$) of methanotrophs during anaerobic methane oxidation[30,31].

By integrating the current with the incubation time, we determined that conductive electron transfer could accumulate extracellular electrons into soil organic matter at a rate of 158 ± 18 μmol $e^-$ $g^{-1}$ soil carbon $day^{-1}$ to stimulate alternative respiration. This rate was eightfold ($N = 3$, $P < 0.01$) faster than that (21 ± 7.7 μmol $e^-$ $g^{-1}$ soil carbon $day^{-1}$) caused by redox-cycling electron transfer. By accumulating electrons to a number of 785 ± 120 μmol $e^-$ $g^{-1}$ soil carbon, which was in the range of the reported maximum electron-accepting capacities (a few hundred to thousand μmol $e^-$ $g^{-1}$ soil carbon) of soil organic matter in several northern peat soils[23,32], we obtained a 24% ($N = 3$, $P = 0.03$) $CH_4$ suppression (blue line in Fig. 1c) in comparison with the pyrogenic carbon-free control treatment. This amount of suppression nearly doubled the amount that was suppressed by the redox-cycling electron transfer and represented an upper limit of the suppressing ability of electron snorkeling effect in the studied peat soil.

Isotopic analyses (Supplementary Method 5) showed that during the redox-cycling electron transfer, a portion of the $CO_2$ and $CH_4$ production was derived from the metabolism of pyrogenic carbon (Fig. 2a and Supplementary Table 1). The easily mineralizable carbon phase (characterized by volatile matter (35–23% w/w) and water extractable carbon (348–211 mg $kg^{-1}$)) is enriched in the pyrogenic carbon that was produced at low (400–500 °C) pyrolysis temperatures[33], which was most likely the available carbon source for microbial metabolization[34]. Although the pyrogenic carbon-derived gas production only accounted for <2% of the total gas production, the metabolism of pyrogenic carbon could induce a shift of the peat soil microbial structure and function[19], which contributed to the suppression of the methanogenesis activity in addition to the redox-cycling induced electron snorkeling process. However, no detectable metabolism of pyrogenic carbon (Fig. 2b and Supplementary Table 2) was observed during the capacitive and conductive electron transfers due to the lack of easily mineralizable carbon phase in the pyrogenic carbon that was produced at high (800 °C) pyrolysis temperature[33].

**Dependency of electron snorkeling on pyrolysis temperatures.** Natural fires generate a highly heterogeneous pyrolysis temperature, which normally covers a range from 400 to 800 °C depending on the size of fires and the types of fuel[35,36]. In some

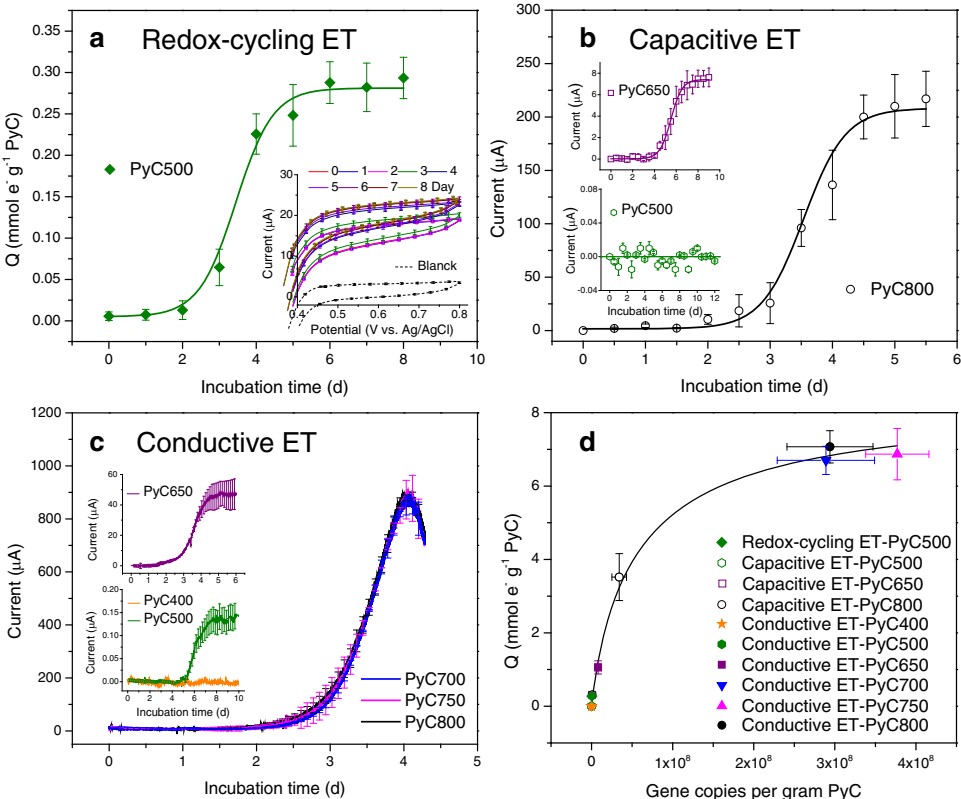

**Fig. 3 Electron transfer kinetics as a function of pyrolysis temperatures determined in the pure-culture (*G. sulfurreducens*) incubations (at 32 °C and ambient light). a** Number of transferred electrons (Q) during the redox-cycling electron transfer (ET) of the pyrogenic carbon functional groups. Redox-cycling ET was tested in the microcosm pure-culture incubations. Pyrogenic carbon was produced at 500 °C (PyC500). Inset: the oxidation current of ferrocyanide after reacting with pyrogenic carbon. **b** Discharging current generated from the capacitive ET through the pyrogenic carbon matrices. The discharging current was the highest current point of each discharging period at a $0.1 \, s^{-1}$ recording frequency. Full discharging current profile could be found at the chronoamperograms in Supplementary Figs. 8–10. **c** Current profiles generated from the conductive ET through the pyrogenic carbon matrices. Both capacitive and conductive ET were tested in the bioelectrochemical pure-culture incubations. Pyrogenic carbon was produced at 400–800 °C (PyC400-PyC800). Day 0 in the x axes in **a**–**c** indicate the time of inoculation of *G. sulfurreducens*. **d** The increased number of transferred electrons (Q) as a function of the increased copy numbers of 16S rRNA genes of *G. sulfurreducens* that was respiring on pyrogenic carbon by using different electron transfer mechanisms. Error bars are s.d. of triplicate measurements. For significant differences, see text. Data are presented as mean values ± s.d.

extreme conditions, such as for the burning of wood core, the temperature can exceed 1000 °C[37]. We investigated the determination of the pyrolysis temperature on the kinetics of electron snorkeling by using the microcosm (Supplementary Method 6 and Supplementary Fig. 3c) and bioelectrochemical (Supplementary Method 7 and Supplementary Fig. 3d) pure-culture (*Geobacter sulfurreducens* strain PCA, without addition of peat soil) incubations. Compared to the multi-phase composition and complexed microbiota in the peat-soil incubations, the pure-culture incubations provided a highly defined system for evaluating the kinetics across a single and identical microbe–pyrogenic carbon interface.

All electron transfer mechanisms demonstrated the electron snorkeling ability in supporting the respiration of *G. sulfurreducens*. This was evidenced by the increased number of transferred electrons as a result of redox-cycling (Fig. 3a and Supplementary Fig. 7) transfer of extracellular electrons and the increased current as results of capacitive (Fig. 3b and Supplementary Figs. 8–10) and conductive (Fig. 3c) transfer of extracellular electrons. In responses to increased electron and current signals, we also detected growing biomass that respired on pyrogenic carbon (Fig. 3d and Supplementary Fig. 11). These results further rationalized the effectiveness of pyrogenic carbon in facilitating extracellular electron transfer and stimulating alternative

respiration as observed in the peat-soil incubations. Both capacitive and conductive electron transfers showed faster kinetics (demonstrated by current slopes within the exponential growth phase of *G. sulfurreducens*) through the carbon matrices that were produced at high-temperature range (700–800 °C) than low to intermediate temperature range (400–650 °C) (Supplementary Fig. 12). This result was consistent with previous abiotic tests that the carbon matrices produced at a high-temperature range possess more ordered polyaromatic carbon ring structures[24] and thus higher electrical capacitance and conductivity[17].

We determined the redox-cycling electron transfer rate of pyrogenic carbon (produced at 500 °C) quinone-hydroquinone couple at $0.16 \pm 0.02 \, mmol \, e^- \, g^{-1}$ pyrogenic carbon day$^{-1}$, based on the increased number of transferred electrons at the *G. sulfurreducens* exponential growth phase (Fig. 3a). This rate was similar ($N = 3$, $P > 0.05$) to the conductive electron transfer rate through the carbon matrices produced at intermediate pyrolysis temperature 650 °C ($0.14 \pm 0.03 \, mmol \, e^- \, g^{-1}$ pyrogenic carbon day$^{-1}$), but slower ($N = 3$, $P < 0.01$) than that of the capacitive and conductive ($0.54 \pm 0.08$ and $3.22 \pm 0.15 \, mmol \, e^- \, g^{-1}$ pyrogenic carbon day$^{-1}$, respectively) electron transfer through the carbon matrices produced at high pyrolysis temperatures (700–800 °C) (Fig. 4). This trend indicated that while the

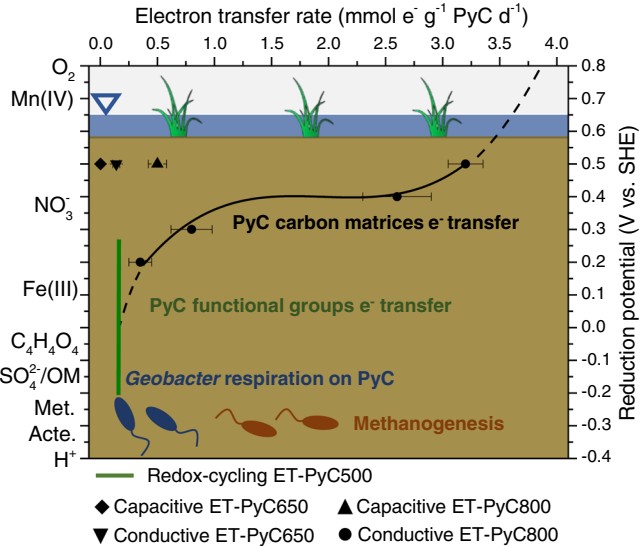

**Fig. 4 Dependency of conductive electron transfer kinetics on environmental terminal electron acceptors.** The conductive electron transfer (ET) rate of the carbon matrices was determined by the linear fit of exponential growth current (Supplementary Fig. 13) that was driven by low to high reduction potentials of terminal electron acceptors. The solid black line is the trend line of measured data and the dashed black line is the extrapolated trend. The reduction potentials and the corresponding terminal electron acceptors listed on the y-axes were adopted from ref. [14,39,59]. In capacitive ET, the reduction potential indicates the potential that drove the capacitance discharge of the carbon matrices. The potential range for driving redox-cycling ET (green line) was derived from the reduction potential of pyrogenic carbon quinone/hydroquinone couple, which was estimated based on direct electrochemical measurement[17] and mediated iron[60] and nitrate[61] reduction. In the figure legend, PyC and the following numbers indicate pyrogenic carbon and pyrolysis temperatures, respectively. SHE in the label of the right y-axis indicates standard hydrogen electrode. OM, Met., and Acte. in the label of the left y-axis indicates organic matter, methanogenesis, and acetogenesis, respectively. Error bars are s.d. of triplicate measurements. Data are presented as mean values ± s.d. Figure 4 was created by authors.

capacitive and conductive electron transfer dominated the electron snorkeling process of the pyrogenic carbon that was produced at high pyrolysis temperatures, the redox-cycling electron transfer became more kinetically preferred in snorkeling electrons for the pyrogenic carbon that was produced at low to intermediate pyrolysis temperatures. Pyrogenic carbon aging potentially results in an attenuation of polyaromatic carbon ring structures in the carbon matrices but an enrichment of surface functional groups[38]. Therefore, the electron transfer mechanisms of pyrogenic carbon are expected to be transitioned from capacitive and conductive electron transfer to redox-cycling electron transfer[25].

Even though the conductive electron transfer through the carbon matrices that were produced at high-temperature range showed rapid electron snorkeling kinetics, its overall performance was highly dependent on terminal electron-accepting conditions. A similar phenomenon was also observed in the peat-soil incubations. We found that dropping the potential gradient by 0.3 V, which is roughly equivalent to the shifting of the terminal electron acceptor from nitrate to iron minerals[14,39], caused a 90% ($N = 3$, $P < 0.01$) decrease of the electron transfer kinetics (Fig. 4 and Supplementary Fig. 13). In contrast, the redox-cycling and capacitive electron transfers were less dependent on terminal electron acceptors due to their respective self-electron acceptance

and storage functions. These functions enable pyrogenic carbon to act as a time-uncoupled intermediate electron acceptor, which sustains electron snorkeling during times when a temporary lack of attached terminal electron acceptors exist[40]. The capacity of the self-electron acceptance and storage can be restored by donating (Fig. 3b) and releasing (Fig. 3c) electrons to available terminal electron acceptors when environmental conditions change. Such condition change can be triggered by, for example, temporary oxygen penetration into peat soils[14,41] or the iron aggregation with pyrogenic carbon[42,43].

## Discussion

Here, we presented a post-fire $CH_4$ suppression phenomenon in a northern peat soil. In line with our findings, recent studies have also reported less methane production in peat soils after forest fire. Davidson et al.[44] found that wildfire imposes more significant impact than water-table fluctuation in reducing methane emission in a treed Canadian peat site. Reduced methane production has also been reported for three rotationally (i.e., biomass) burnt peatland sites (see: http://peatland-es-uk.york.ac.uk/field-measurements/ghg-emissions). By showing the links between fire-derived pyrogenic carbon and methanogens' activity, this study provides a new insight in explaining the mechanisms of fire-induced methane reduction. In contrast to the traditional ideas that rationalize the legacy effect of pyrogenic carbon mainly from the perspective of its persistence[45,46], the $CH_4$ suppression observed in this study was a result of strong redox and bioelectrochemical interactions between pyrogenic carbon and microbial respiration, which provides a new direction of re-evaluating the legacy effect of pyrogenic carbon.

Previous studies in anaerobic digesters[47] or co-culture (*Geobacter and Methanogen*) incubations[48] have reported increased $CH_4$ production after addition of carbonaceous materials (e.g., activated carbon). However, these engineered systems are different from naturally occurring peat soils. To achieve high $CH_4$ yield, there are usually very limited amount of alternative electron acceptors available in anaerobic digestors and co-culture incubations, which creates a thermodynamically favorable environment for methanogens' metabolism (i.e., electron flows derived from acetate oxidation are preferentially accepted by carbon dioxide through methanogenesis process). In contrast, peat soils generally contain a variety of alternative electron acceptors, for instance the iron oxides[13] and quinone molecules of soil organic matter[14,49]. It has been shown that continuous $CH_4$ production can only start after the electron-accepting capacity of soil organic matter being consumed to an extent that drives a slower alternative respiration than methanogenesis[23]. We observed similar phenomenon in the pyrogenic carbon-free control treatment that slower alternative respiration appeared after evolution of $CH_4$ production (inset chart in Fig. 1b). Addition of pyrogenic carbon, however, could further stimulate alternative respiration and reduce $CH_4$ production by facilitating extracellular electron transfer to a larger electron-accepting pool of soil organic matter (Fig. 1b). During our experimental period of 9 days, we did not observe the depletion of the electron-accepting capacity of soil organic matter, as the number of accepted electrons was still increasing by the end of experiments. Therefore, the electron snorkeling process was expected to be able to maintain throughout a longer period.

Both pyrogenic carbon functional groups and carbon matrices contributed to the stimulation of alternative respiration by employing redox-cycling, capacitive, and conductive electron transfer mechanisms. This multi-electron transfer mechanism allowed pyrogenic carbon to process the electron snorkeling across a wide range of naturally occurring fire/pyrolysis

temperatures (Fig. 3a–c). Given the fact that pyrogenic carbon is an intrinsic part of peat soils[8], the 13-24 % reduction in $CH_4$ production is potentially responsible for a considerable cooling effect to climate and merits a consideration in the global carbon budget estimation. However, it should be noted that the $CH_4$ suppression power of pyrogenic carbon estimated in this study applies to the situation with deposited pyrogenic carbon from aboveground vegetation fires in unburnt peat soil. It should not be confused with the carbon losses from hot fires burning into the peat and causing significant amounts of net carbon emissions[50].

Increased accumulation of pyrogenic carbon is expected as a result of more frequent forest fires[51]. Higher amount of pyrogenic carbon caused greater activity of alternative respiration (Supplementary Fig. 14). However, the number of snorkeled electrons did not show an accelerated increase in response to the increased biomass in alternative respiration, instead, the increasing rate decelerated and eventually leveled out (Fig. 3d). This response pattern resembles the Michaelis–Menten behavior in enzymatic chemistry, which reflects the catalytic nature of the electron snorkeling process. On the other hand, the decelerated increasing rate implies that a functioning ceiling existed in the electron snorkeling process, which limits further $CH_4$ suppression even though the input of pyrogenic carbon is sufficient. In addition to pyrogenic carbon accumulation, other factors like vegetation change[52], fire severity[53], and temperature variation[54] have also been shown to influence carbon fluxes in fire-altered soils. Therefore, we suggest future studies should focus on large-scale field test to track to what limit pyrogenic carbon can suppress peat $CH_4$ production and on quantifying how much contribution this suppression can make among other factors to offset combustion losses from peatland vegetation emission. Such studies will potentially lead to a paradigm shift of the pyrogenic carbon legacy effect from passive carbon sequestration to active mitigation of greenhouse gas emissions from the peatland soils.

## Methods

**Pyrogenic carbon samples.** Pyrogenic carbon samples were produced under controlled conditions in the laboratory by anoxic pyrolysis of woody biomass from shrub willow (*Salix viminalis x S. miyabeana*). We used shrub willow for the production of pyrogenic carbon because it is a common plant in shrubby peatland and its electron transfer mechanisms can be representative of woody biomass-derived pyrogenic carbon due to the similar formation and composition of carbon matrices and functional groups during the pyrolysis of woody materials[16,17,24]. We applied pyrolysis temperatures of 400–800 °C, which covered the temperature range of naturally occurring peatland fires[35,36] and created a spectrum of aromaticity and surface functionality[17,24,25] that represented the structural transition of naturally occurring pyrogenic carbon[55]. We used a dwell time of 60 min for all pyrolysis temperatures. Before pyrolyzing, the biomass carbon had been labeled in growth chambers using $^{13}C$ labeled $CO_2$ as the source for willow growth (a detailed labeling process can be found in ref. [33]). The bulk $\delta^{13}C$ in pyrogenic carbon after biomass pyrolysis was on average 774 ± 2.3‰ (vs. VPDB). More information on the physicochemical properties (carbon content, elemental composition, pH, etc.) of pyrogenic carbon can be found in our previous study[33]. Even though lab-made pyrogenic carbon is structurally more homogeneous than naturally occurring pyrogenic carbon produced by vegetation fires, the revealed electron transfer mechanisms are applicable to both cases as they follow the same carbonization and functionalization trend in response to the change of pyrolysis temperatures[55].

**Peat soil sample.** We sampled the peat soil from the Mclean Bog located in Dryden, New York (42°30′ N, 76°30′ W). The Mclean Bog is a national natural landmark with known methanogenesis activity and $CH_4$ production. The peat soil is saturated to close to the peat surface, except during extreme drought periods. We collected soil samples in June under saturated condition. The samples were collected from the surface and reached a depth of ~10–15 cm and contained 52% total carbon and 2.1% total nitrogen (dry weight percentage). Detailed description of the peat soil was given in Supplementary Method 1. For all peat soil incubations, 10 g wet soil was suspended with deoxygenated and deionized water to reach a final volume of 30 ml. The final pH of the soil suspension was 4.5, and all pyrogenic carbon had been adjusted to this pH prior to incubation. pH adjustment was conducted with sodium hydroxide or hydrogen chloride in water media

(containing 10% ethanol to reduce the surface tension of pyrogenic carbon). The $CH_4$ production was very slow (<1 μmol g$^{-1}$ soil carbon day$^{-1}$) in the first few days of incubation. It started to rise after day 6. The $CH_4$ production rate, thereafter, was in the range of 4.7–5.9 μmol g$^{-1}$ soil carbon day$^{-1}$ (or 0.08–0.1 mmol L$^{-1}$ soil suspension day$^{-1}$) (Supplementary Fig. 2), which was in agreement with previously reported rates (0.02–0.1 mmol L$^{-1}$ soil suspension day$^{-1}$) from the same peat soil[22,56]. Accordingly, pyrogenic carbon was added to the peat soil after 6 days of pre-incubation.

**Pure-culture inoculum.** The bacterial pure-culture *G. sulfurreducens* strain PCA (1 mL of stock culture at 0.1 OD) was inoculated in all pure-culture incubations. From all electron-donating bacteria, we chose *G. sulfurreducens* because of its mediator-free electron transfer mechanisms that facilitated the mechanism study by only crossing one electron transfer interface (i.e., bacteria-pyrogenic carbon interface instead of bacteria-mediator-pyrogenic carbon interfaces) and avoided potential interference caused by the mass transport of mediators. Further, *G. sulfurreducens* is an important and abundant alternative-respiring bacterium that coexists with methanogens in many anaerobic environments[57]. The average abundance of *Geobacter* species accounted for 1% of 16S rRNA gene sequences, which dominated the alternative respiratory bacteria in the studied peat soil (Supplementary Fig. 1).

**Microcosm and bioelectrochemical incubations.** The microcosm and bioelectrochemical incubations were performed to investigate the electron snorkeling induced by the pyrogenic carbon functional groups and carbon matrices, respectively. For studying electron snorkeling of pyrogenic carbon functional groups, we used small pyrogenic carbon particles to reach a homogeneous mix with peat soil and electrochemical quantification of electron-accepting capacities of pyrogenic carbon and peat soil mixture. We prepared small pyrogenic carbon particles by following a published ball milling method, which yielded a particle size of ~40 μm[16]. We used a large particle size (~1.5 cm) during the investigation of pyrogenic carbon electron snorkeling to facilitate the incorporation of pyrogenic carbon (as a working electrode) in the bioelectrochemical circuit. This range of pyrogenic carbon particle size falls into the sizes of pyrogenic carbon that has been found in soils[58].

We performed peat-soil and pure-culture incubations in parallel together with pyrogenic carbon in microcosm and bioelectrochemical incubations (at 32 °C). The application rate was 0, 0.03, and 3 mg pyrogenic carbon g$^{-1}$ soil in the microcosm peat-soil incubations and 0 and 1 mg pyrogenic carbon mL$^{-1}$ growth medium in the microcosm pure-culture incubations. The application rate was 0 and 10 mg pyrogenic carbon g$^{-1}$ soil and 0 and 6.7 mg pyrogenic carbon mL$^{-1}$ growth medium for bioelectrochemical peat-soil and pure-culture incubations, respectively. We monitored daily production rates of $CO_2$ and $CH_4$ and $\delta^{13}C$ partitioning in the gas phase of all bioelectrochemical and microcosm incubations, using a Picarro stable isotope analyzer (G2201-I, Santa Clara, CA, USA). Detailed information about the design, operation, and data acquisition of microcosm and bioelectrochemical incubations could be found in Supplementary Method 2–4 and 6–7. Statistical analyses of the data were performed using Excel and Origin version 8.5 with one-way ANOVA and least significant difference for comparison of treatment means with a significance threshold of $P < 0.05$ ($n = 3$). Sub-samples for triplicated incubations were from the same peat soil.

**Reporting summary.** Further information on research design is available in the Nature Research Reporting Summary linked to this article.

## Data availability
The data supporting the findings of this study are available within the article and its Supplementary Information files. Source data for figures are provided with this paper.

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

## Acknowledgements

This research was supported by NSF-BREAD (grant number IOS-0965336), USDA NIFA Carbon Cycles (2014-67003-22069), and the Alexander von Humboldt Foundation in the framework of the Alexander von Humboldt Professorship endowed by the Federal Ministry of Education and Research in Germany. Any opinions, findings and conclusions, or recommendations expressed in this material are those of the authors and do not necessarily reflect the views of the donors.

## Author contributions

T.S., J.L., and L.T.A. planned the research, T.S. conducted the bioelectrochemical and microcosm incubation experiments, laboratory analysis, and data analysis. T.S. wrote the manuscript in close collaboration with L.T.A. J.J.L.G. assisted with the bioelectrochemical measurements and interpretation and qPCR analysis. J.D.S. and J.B.Y. provided the peat soil samples and performed the physicochemical and genomic analyses. A.E. assisted the setup of incubation and gas and isotopic analysis. All authors edited the manuscript.

## Funding

## Competing interests

The authors declare no competing interests.
