## [Peer Review File · Nature Communications]

Editorial Note: This manuscript has been previously reviewed at another journal that is not operating a transparent peer review scheme. This document only contains reviewer comments and rebuttal letters for versions considered at Nature Communications. Mentions of prior referee reports have been redacted.

REVIEWER COMMENTS

Reviewer #1 (Remarks to the Author):

GENERAL COMMENTS

This is a very novel and thorough laboratory experiment that uncovers potential mechanisms by which pyrogenic carbon could be decreasing CH₄ emissions in peat soils. In a previous version, submitted to Nature Climate Change, the authors overstated the environmental implications of their findings; however, in this current version, they have done a fine job in putting their results in the right context. I am still not sure if these findings are sufficiently relevant for the readership of Nat. Com., but that is something that the editors should decide on. I have just some comments below and in the annotated manuscript attached:

- Considering the broad readership of this journal it would be very useful to have a figure, maybe in the introduction, with a diagram showing the different processes (methanogenesis, alternative respiration, etc) in peat soils, with main substrates and microbial groups involved, and the potential role of PyC. This would be very useful for the readers to get the overall idea and would lay excellent basis for the authors to present their hypotheses (L71-75).
- Regarding the 3 mechanisms for CH₄ production reduction (L111-114), it is not clear if the first one was only observed for 400-500PyC (Fig. 1) and the second one only for 800PyC (Fig. 2), or if these were only examples, or they were the PyC types where these mechanisms were stronger. This needs to be clarified.
- In the Discussion, the authors should briefly mention the limitations of having such a short incubation time (9 days) and whether the observed trends are expected to be maintained over longer periods of time.
- In the Methods section the authors explain that by using 400-800 degs during PyC production, they "created a spectrum of aromaticity and surface functionality"; however, nowhere in manuscript, the figures about aromaticity and surface functionality are shown.

MINOR COMMENTS

- Figure S1: not clear what "We sampled the soil at 10-15 cm" means.
- See other specific comments in the annotated manuscript attached.

Reviewer #3 (Remarks to the Author):

I have to congratulate the authors in addressing the raised issues by the three reviewers. The rebuttal provides a convincing case, clarifying further issues as well as highlighting corresponding changes made to the manuscript. This also includes the title and abstract, which now clearly state the artificial conditions and the limitations of this work (highlighting clearly in the manuscript conclusions the need for further field based research).

I find all the responses clear and sound. However, some minor things I would like to see addressed are:

In track changed MS:

- 1) page 5: access the upper (suggest replace limit with potential)
- 2) page 8: redox cycling of pyrogenic (suggest replace show with observed)
- 3) Discussion page 13:
Less (replace with Reduced) methane production has also...

has also been found in several burnt (replace with has also been reported for three rotationally (i.e. biomass) burnt peatland sites (Heinemeyer et al., 2019, see: webpage is OK as is)

Heinemeyer A., Vallack H.W., Morton P.A., Pateman R., Dytham C., Ineson P., McClean C., Bristow C. and Pearce-Higgins J.W. (2019) Restoration of heather-dominated blanket bog vegetation on grouse moors for biodiversity, carbon storage, greenhouse gas emissions and water regulation: comparing burning to alternative mowing and uncut management. Final Report to Defra on Project BD5104, Stockholm Environment Institute at the University of York, York, UK.

page 14: move 'in engineered ...' AFTER 'Previous studies INSERT ...' AND remove the) before the end.

page 16:

replace 'wooden' with woody (3 times)

insert 'a' before (detailed labeling...)

replace 'at' with in (June under saturated condition.)

page 18:

replace 'for' with to achieve a (homogeneous mix with...)

add 'a' before large particle size (~1.5 cm)...

add 'a' before (...working electrode)

page 19:

replace 'run' with performed

add 'a significance threshold of ' before $P < 0.05$ (n=3)

Graphs: Fig. 4 add the before the new text (of THE left y ... & of THE right y...)

You now mention the P-values in the text, but could you maybe add "for significant differences see text" to the graph legends - but ideally show some *** etc in the actual graph and explain it in the legend.

Reviewer #4 (Remarks to the Author):

The goal of this study was to illustrate that pyrogenic carbon (char) from forest fires in peatlands can act as an "electron snorkel" to promote "alternative respiration" and reduce methanogenesis. The authors used peat soil samples collected from Mclean Bog, NY, and whole and ground chars prepared in the lab through pyrolysis at 400-800 °C of shrub willow grown on ^{13}C to perform four types of experiments: batch incubations with 400/500 °C chars and either soil or *Geobacter sulfurreducens*, and electrochemical bioreactors with either peat soil and 800 °C char, or *G. sulfurreducens* and chars made at 400-800 °C. For the electrochemical experiments, a potential of +0.5V (vs. SHE) was applied to the char (used as working electrode) either twice a day for 30 min or continuously. All experiments lasted 9 days.

The authors propose that 400-800 °C char can enhance microbial respiration to CO_2 and suppress CH_4 production, through three mechanisms: (1) redox cycling of quinone groups in soil organic matter (SOM), (2) capacitive electron transfer through solid carbon matrices, and (3) conductive electron

transfer through solid carbon matrices.

The ability of humic substances (HS, including SOM) to act as terminal electron acceptors (EA) is well-documented, and that redox cycling of HS/SOM can promote anaerobic respiration and suppress methane formation has been proposed (in 2014 if not earlier). For HS, mechanism (1) is the only mechanism involved.

The authors' central hypothesis is that, char can channel, or snorkel, electron to SOM to promote this process, through all three mechanisms above. Based on their data, I agree only partially. The data (e.g., Figure S4) show that chars, especially chars prepared at 400-500 oC, can serve as EA, just like SOM. And if SOM is present in great excess (as in their experiments and in peatland soils), through redox equilibrium chars can shuttle electron from microbes to SOM over repeated cycles. (By the way, SOM can do exactly the same.) This part is fine, though I do have a few questions/comments as discussed below. The 2nd and 3rd mechanisms, however, I believe are experimental artifacts that likely have little relevance in nature.

Overall, the manuscript contains experimental evidence for a new/not yet widely recognized role of forest fire chars in anaerobic microbial processes that have potential climate implications. For that reason, the manuscript may be eventually publishable. However, significant rewriting and reinterpretation of some of the data, preferably with a more focused hypothesis, a narrower scope, and additional measurements, would be necessary for the manuscript to be considered further.

Additional Comments:

1. Do not use the term "electron snorkel". Mechanistically, its meaning is unclear. Be specific and say electron transfer, electron conduction, or redox reaction. Using ill-defined terms only obfuscates, not clarifies. Do not use "catalyzing" (p.4). None of the 3 proposed mechanisms involve char as a catalyst. What is "alternative respiration"? Alternative to what? If it is O₂, change it to "anaerobic respiration".

2. The pyrogenic carbon matrices cannot possess a potential range of 1.5 V (p.4). The potential gap between the O₂/H₂O and H⁺/H₂ redox couples is only 1.22 V (at any pH). A carbon that possesses a potential range of 1.5 V would necessarily reduce or oxidize water to H₂ or O₂, respectively.

3. Figure 1a,b: The data suggest the added char reduced CH₄ formation, probably by serving as EA. However, a quantitative electron balance was not obtained. Also, the electron donating and accepting capacities (EDC and EAC) of the chars were not measured. Based on the lower mass of CH₄ formed, one can calculate how much electron had been re-directed to respiration. How does the 147 μmol e⁻/g soil recovered (p.6) compare to the EAC of the chars (so that we would know the relative importance of chars and SOM as EA)? Additional measurements, calculations, and data analysis are needed for 400/500 oC char results.

It is clear from Figures S4 and S7 that chars prepared at 400/500 oC (as well as soil/SOM and cells of *G. sulfurreducens*) contain positive EDC. If the authors had pre-oxidized the chars with dissolved oxygen (to remove the EDC and maximize the EAC) before use, as has been done with humic materials (see, e.g., work from Sander's group at ETH), then the data would be less ambiguous and the role of chars as EA would likely be more pronounced.

4. Figure 1c,d, and Figures S6,8,9,10,11: It is not clear how the "capacitive" electron transfer mechanism works, how the "capacitive" and "conductive" mechanisms differ, and whether either mechanism is relevant in actual peatland soils. The three questions are discussed below.

First, by "capacitive", do the authors mean accumulation of opposite charges at spatially separate locations of a single piece of char? I do not see how this would happen, either in reactors or in field soils. A piece of reduced char in a soil slurry or reactor would be at redox equilibrium with the

surrounding solution and SOM. Since there is only one redox potential dictated by the local redox equilibrium, there would be no capacitance across a solid carbon matrix (e.g., electron exchange with aqueous SOM would remove any capacitive potential). Did the authors measure a non-zero potential across the working electrode (i.e., solid char), for example, in Figure S3b? If they did, which would support their capacitive mechanism hypothesis, please show the data.

If by "capacitive" the authors simply suggest accumulation of excess electrons in chars that contain crystalline/graphitic regions with fermi levels to accommodate electrons of different potentials, then a separate measurement should have been made to demonstrate the existence of such regions and to determine the electron holding capacity of 800 oC char. In this case, though, the char is not acting as a capacitor, and the term "capacitive" should not be used.

In any case, please clarify what is meant by "capacitive electron transfer" and how it would work exactly, and provide supporting data if available.

Second, it is unclear how the "capacitive" and "conductive" electron transfer mechanisms are different. Experimentally, an oxidizing (+0.5V) potential was applied to char in both systems, but only for 30 min twice a day for the former and continuously for the latter. As one would anticipate, a positive potential would favor/select for exo-electrogenic microbes, such as *G. sulfurreducens*, promoting oxidation of acetate and other substrates to CO₂, and minimizing fermentation and methanogenesis. In addition, the higher the potential, the longer it was applied, and the more conductive the char was, the more substrate oxidation and cell growth would occur. These are consistent with Figure 1c and 1d, and Figures S8, S9, S10, S11, S12 and S13.

Thus, by simply applying the same potential for different durations, it is not clear whether the authors were testing two different mechanisms. If the capacitive and conductive mechanisms really are different (in theory, not just operationally), please explain how exactly.

Third, applying an oxidizing potential would not only support substrate oxidation and growth of *G. sulfurreducens* and other electrode-respiring bacteria, but also (abiotically) oxidize any electrode-active reductants. SOM happens to be electrode-active and would adsorb favorably to graphitic carbons, such as high temperature chars. How do we know the applied potential did not periodically or continuously re-oxidize hydroquinones and catechols (which would contribute to the observed currents) in SOM sorbed to char? This would re-generate quinones that could then serve as EA to support further respiration. It is not clear to what extent this occurred in electrochemical reactors. Hence, the "capacitive" and "conductive" mechanisms may be, at least in part, the redox cycling mechanism in disguise.

As noted above, both SOM and *G. sulfurreducens* have positive EDC (Figures S4 and S7). The applied potential would pull electron out of attached *G. sulfurreducens* cells and SOM in contact with the char, in addition to acetate (through microbial metabolism). Therefore, the observed currents most likely exaggerate the importance of the conduction mechanism. Note that (quinones in) SOM is the dominant and ultimate EA in peatland soils. Char conductivity itself cannot support respiration without SOM as EA. In the absence of an imposed potential, I don't see how char conductivity would play a significant role in situ. (Unless quinones in sorbed SOM can be respired directly through conduction, which this study does not show.)

5. Figure 2: What prompted the use of ¹³C? All three proposed mechanisms would not produce ¹³CO₂ or ¹³CH₄, so it is unclear why ¹³C-enriched chars would help. How is biodegradability of chars relevant to any of the hypotheses? Please explain the rationale. It makes sense that the lower temperature chars were more labile than 800 oC char; however, biodegradation of 400/500 oC chars would likely yield both ¹³CO₂ and ¹³CH₄ (because the redox state of carbon would change from ~zero to +4 and -4) rather than favoring CO₂ over CH₄.

6. SI p.10: Should be "Eq. S4 + Eq. S5" instead of "Eq. S3 + Eq. S4".

7. Method S3 (SI p.5): I would be careful with the assumption that pyrogenic carbon produced at 400-500 oC transferred electron through redox cycling involving quinone functional groups, whereas carbon produced at 800 oC had condensed carbon structure and therefore transferred electron through conductive solid carbon matrices.

I agree 800 oC char would be more conductive and 500 oC char would contain more quinones and other redox functional groups, but the O/C and H/C ratios of these carbons differed only by a factor of 2 (SI p.5). The EDC and EAC of these chars should have been measured and compared to assess the contributions of functional groups to the overall electron transfer. Conversely, it has been shown that small (3-6 nm) graphene islands can form at pyrolysis temperatures as low as 500 oC, which would enable electron transfer from, e.g., conductive pili and extrusions of exoelectrogenic cells to adsorbed soil organic matter, even though the bulk solid of 500 oC carbon was poorly conductive.

Responses to reviewers' comments

We are grateful to the reviewers for their positive feedbacks to our first revision. We are glad that we were able to address most of the reviewers' concerns. We also thank the new reviewer for providing new insightful comments to further improve the quality of the manuscript. We have revised the manuscript accordingly. In addition to a clean version of the thoroughly revised manuscript, any changes in response to reviewers' comments have also been highlighted in the change-tracked version of the revised manuscript main text and supplementary information. A clean version of all revised documents has also been provided to the reviewers for easier reading after revision. Below is a point by point response to the reviewers' comments. All references cited in this response letter are given at the end of the letter. Texts highlighted in blue in this letter are also the revised texts in the manuscript. We show them here for the convenience of reviewing process.

Reviewers' comments:

Reviewer #1

General comments:

This is a very novel and thorough laboratory experiment that uncovers potential mechanisms by which pyrogenic carbon could be decreasing CH₄ emissions in peat soils. In a previous version, submitted to Nature Climate Change, the authors overstated the environmental implications of their findings; however, in this current version, they have done a fine job in putting their results in the right context. I am still not sure if these findings are sufficiently relevant for the readership of Nat. Com., but that is something that the editors should decide on. I have just some comments below and in the annotated manuscript attached.

Author response: We thank the reviewer for appreciating the novelty and comprehensiveness of this study. We are also glad that we were able to address most of the reviewer's concerns in the first round of revision. In this second revision, we, again, carefully considered every comment given by the reviewer and improved the manuscript accordingly.

Comment 1. Considering the broad readership of this journal it would be very useful to have a figure, maybe in the introduction, with a diagram showing the different processes (methanogenesis, alternative respiration, etc) in peat soils, with main substrates and microbial groups involved, and the potential role of PyC. This would be very useful for the readers to get the overall idea and would lay excellent basis for the authors to present their hypotheses (L71-75).

Author response: We fully agree with the reviewer that a schematic diagram which outlines different processes and microbes can be useful for broad readers to understand our study. This is also why we provided a **Figure 4** in the manuscript. The reason we put this figure in the Results section instead of the Introduction section is to incorporate more quantitative results into qualitative demonstration. This way, the readers can on one hand easily understand the overall concept of our study and on the other hand use this figure as a guidance to assess the predominant respiration pathway at conditions of different electron

donor and acceptor combinations. However, based on the reviewer's comment, we have further improved **Figure 4** by exchanging the left and right Y axes to highlight the main substrates that are competed between methanogenesis and alternative respiration. We have also clarified the dominant microbes in the figure. We thank the reviewer for this advice. We have added the revised figure in the manuscript. We also present the figure below to facilitate the reviewing process.

Comment 2. Regarding the 3 mechanisms for CH₄ production reduction (L111-114), it is not clear if the first one was only observed for 400-500PyC (Fig. 1) and the second one only for 800PyC (Fig. 2), or if these were only examples, or they were the PyC types where these mechanisms were stronger. This needs to be clarified.

Author response: These 3 electron transfer mechanisms exist in the PyC produced at all temperatures. However, at low temperature range (400-500°C), as the reviewer postulated, the redox-cycling electron transfer mechanism is kinetically much faster than conductive and capacitive electron transfer mechanisms, therefore dominates the electron snorkeling process. A reversed trend has been found at high temperature range. We have now clarified these kinetic features in the manuscript and also presented the changes below. We thank the reviewer for pointing this out.

“We found that three electron transfer mechanisms contributed to electron snorkeling of pyrogenic carbon and caused significant decreases in CH₄ production: (1) redox-cycling electron transfer by the quinone and hydroquinone functional groups; (2) capacitive

electron transfer through the carbon matrices; and (3) conductive electron transfer through the carbon matrices. While redox-cycling electron transfer was the kinetically preferred mechanism for the electron snorkeling of pyrogenic carbon that was produced at low temperature range, capacitive and conductive electron transfer mechanisms dominated the electron snorkeling of pyrogenic carbon that was produced at high temperature range (Figure 1 and SI Figure S2)”.

Comment 3. In the Discussion, the authors should briefly mention the limitations of having such a short incubation time (9 days) and whether the observed trends are expected to be maintained over longer periods of time.

Author response: The electron snorkeling effect of PyC largely depends on the electron accepting capacity of extracellular electron acceptors (e.g., soil organic matter in this study). After the accepting capacity is being saturated, electron snorkeling process will be ceased. In our 9 days experiment, we did not observe the saturation phenomena as the electron accepting capacity was still increasing by the end of experiments. Therefore, the electron snorkeling process was expected to be able to maintain over longer periods of time. We have added this discussion in the manuscript. The added text has also given below.

“Addition of pyrogenic carbon, however, could further stimulate alternative respiration and reduce CH₄ production by facilitating extracellular electron transfer to a larger electron accepting pool of soil organic matter (Figure 1b). During our nine-day experimental period, we did not observe the depletion of the electron accepting capacity of soil organic matter, as the number of accepted electrons was still increasing by the end of experiments. Therefore, the electron snorkeling process was expected to be able to maintain throughout a longer period”.

Comment 4. In the Methods section the authors explain that by using 400-800 degs during PyC production, they “created a spectrum of aromaticity and surface functionality”; however, nowhere in manuscript, the figures about aromaticity and surface functionality are shown.

Author response: We concluded the spectrum of aromaticity and functionality based not on this study but on our (Sun et al., 2017; Sun et al., 2018) and other (Keiluweit et al., 2010) previous studies that are focused on the characterization of PyC physicochemical properties. In these studies, the composition and transition of aromaticity and functionality as a function of pyrolysis temperatures have been characterized by a variety of techniques, mainly including fourier-transform infrared spectroscopy and synchrotron and electron energy loss spectroscopy. We have now added these literature references in the manuscript (as shown below). We thank the reviewer for pointing this out to us.

“...created a spectrum of aromaticity and surface functionality (Keiluweit et al., 2010; Sun et al., 2017; Sun et al., 2018)...”

Minor comments:

Comment 5. Figure S1: not clear what “We sampled the soil at 10-15 cm” means.

Author response: We meant we sampled the soil from the bog surface and reached a depth of 10-15 cm. However, this information is irrelevant to this figure. We have deleted it. We thank the reviewer for pointing this out.

Comment 6. See other specific comments in the annotated manuscript attached.

Author response: The editor wrote us that this annotated manuscript had been lost in the uploading process by the reviewer. From our side we could not see the annotated comments in the uploaded document.

Reviewer #3

I have to congratulate the authors in addressing the raised issues by the three reviewers. The rebuttal provides a convincing case, clarifying further issues as well as highlighting corresponding changes made to the manuscript. This also includes the title and abstract, which now clearly state the artificial conditions and the limitations of this work (highlighting clearly in the manuscript conclusions the need for further field based research).

Author response: We thank the reviewer for the insightful and constructive comments that helped us to improve the first version of the manuscript. We are also glad that the reviewer appreciated our revision and addressing of raised issues.

I find all the responses clear and sound. However, some minor things I would like to see addressed are:

Comment 1. In track changed MS: page 5: access the upper (suggest replace limit with potential)

Author response: The word “limit” has been replaced by “*potential*”.

Comment 2. page 8: redox cycling of pyrogenic (suggest replace show with observed)

Author response: The word “show” has been replaced by “*observe*”.

Comment 3. Discussion page 13: Less (replace with Reduced) methane production has also... has also been found in several burnt (replace with has also been reported for three rotationally (i.e. biomass) burnt peatland sites (Heinemeyer et al., 2019, see: webpage is OK as is)

Author response: We have replaced “less” with “*reduced*”. We have also modified the sentence to “*Reduced methane production has also been reported for three rotationally (i.e. biomass) burnt peatland sites (Heinemeyer et al., 2019)*”.

Comment 4. page 14: move 'in engineered ...' AFTER 'Previous studies INSERT ...' AND remove the) before the end.

Author response: Now the new sentence is “*Previous studies in anaerobic digesters or co-culture (Geobacter and Methanogen) incubations have also reported increased CH₄ production after addition of carbonaceous materials (e.g., activated carbon)*”.

Comment 5. page 16: replace 'wooden' with woody (3 times); insert 'a' before (detailed labeling...); replace 'at' with in (June under saturated condition.)

Author response: All have been revised accordingly.

Comment 6. page 18: replace 'for' with to achieve a (homogeneous mix with...); add 'a' before large particle size (~1.5 cm)...; add 'a' before (...working electrode)

Author response: All have been revised accordingly.

Comment 7. page 19: replace 'run' with performed; add 'a significance threshold of ' before $P < 0.05$ (n=3); Graphs: Fig. 4 add the before the new text (of THE left y ... & of THE right y...)

Author response: All have been revised accordingly.

Comment 8. You now mention the P-values in the text, but could you maybe add "for significant differences see text" to the graph legends - but ideally show some *** etc in the actual graph and explain it in the legend.

Author response: We have added the *"For significant differences, see text"* note in the legends of Figure 1-3. We thank the reviewer for this idea.

We also tried to make *** marks in figures to denote significant differences. However, these marks made the figures a bit too crowded to read, and thus did not work the way as expected.

Reviewer #4

General comments:

The goal of this study was to illustrate that pyrogenic carbon (char) from forest fires in peatlands can act as an "electron snorkel" to promote "alternative respiration" and reduce methanogenesis. The authors used peat soil samples collected from Mclean Bog, NY, and whole and ground chars prepared in the lab through pyrolysis at 400-800 oC of shrub willow grown on $^{13}\text{CO}_2$ to perform four types of experiments: batch incubations with 400/500 oC chars and either soil or *Geobacter sulfurreducens*, and electrochemical bioreactors with either peat soil and 800 oC char, or *G. sulfurreducens* and chars made at 400-800 oC. For the electrochemical experiments, a potential of +0.5V (vs. SHE) was applied to the char (used as working electrode) either twice a day for 30 min or continuously. All experiments lasted 9 days.

The authors propose that 400-800 oC char can enhance microbial respiration to CO_2 and suppress CH_4 production, through three mechanisms: (1) redox cycling of quinone groups in soil organic matter (SOM), (2) capacitive electron transfer through solid carbon matrices, and (3) conductive electron transfer through solid carbon matrices.

The ability of humic substances (HS, including SOM) to act as terminal electron acceptors (EA) is well-documented, and that redox cycling of HS/SOM can promote anaerobic respiration and

suppress methane formation has been proposed (in 2014 if not earlier). For HS, mechanism (1) is the only mechanism involved.

The authors' central hypothesis is that, char can channel, or snorkel, electron to SOM to promote this process, through all three mechanisms above. Based on their data, I agree only partially. The data (e.g., Figure S4) show that chars, especially chars prepared at 400-500 °C, can serve as EA, just like SOM. And if SOM is present in great excess (as in their experiments and in peatland soils), through redox equilibrium chars can shuttle electron from microbes to SOM over repeated cycles. (By the way, SOM can do exactly the same.) This part is fine, though I do have a few questions/comments as discussed below. The 2nd and 3rd mechanisms, however, I believe are experimental artifacts that likely have little relevance in nature.

Overall, the manuscript contains experimental evidence for a new/not yet widely recognized role of forest fire chars in anaerobic microbial processes that have potential climate implications. For that reason, the manuscript may be eventually publishable. However, significant rewriting and reinterpretation of some of the data, preferably with a more focused hypothesis, a narrower scope, and additional measurements, would be necessary for the manuscript to be considered further.

Author response: We appreciate the reviewer for a careful reading of our manuscript. The understanding of the goals, sample preparation procedure, experimental methodology, and major findings of this study are in general in line with what we wanted to deliver. We value this opportunity to discuss and exchange ideas with the reviewer and finally improve the quality of the manuscript.

We agree with the reviewer that SOM has been widely shown to be able to accept electrons and thus contribute to suppressing methane emission. We have clarified these discoveries and cited relevant literatures in both main manuscript and supplementary information. In contrast, the major new finding given by this study was the function of pyrogenic carbon in transferring electrons and suppressing methane production, by actively interacting with SOM and extracellular electron transfer microbes. Increased accumulation of pyrogenic carbon has been widely indicated as a result of more frequent and severe forest fires (Flannigan et al., 2006). A better understanding of the interaction mechanisms among pyrogenic carbon, SOM, and microbes can help to precisely predict the role of pyrogenic carbon in regulating gas emissions in peatland soils and provide new insights in explaining the observed reduction of methane emission after forest fires.

Specific comments:

Comment 1. Do not use the term "electron snorkel". Mechanistically, its meaning is unclear. Be specific and say electron transfer, electron conduction, or redox reaction. Using ill-defined terms only obfuscates, not clarifies. Do not use "catalyzing" (p.4). None of the 3 proposed mechanisms involve char as a catalyst. What is "alternative respiration"? Alternative to what? If it is O₂, change it to "anaerobic respiration".

Author response: We thank the reviewer for this comment. We agree with the reviewer that a well-defined term can help to deliver messages more clearly and straightforwardly. This is also why we chose "electron snorkel", which is a well-defined and commonly used term.

Electron snorkel refers to a conductive or redox-active material that is able to facilitate electron transfers from terminal electron donor to acceptor. Examples of using electron snorkel as a term can be found in (Cruz Viggi et al., 2015; Hoareau et al., 2019; Mitov et al., 2021; Yang et al., 2013). In other words, we did not invent this term, we just adopted it to characterize pyrogenic carbon and highlight its function in facilitating electron transfers.

We have replaced “catalyze” with “*facilitate*” through the manuscript.

Yes, we meant alternative to O₂. We have clarified this point in the manuscript. In addition, anaerobic respiration is a general term, which also includes the respiration of methanogens. To be more specific, we used alternative respiration to refer to the anaerobic respiration that uses alternative electron acceptors. We defined alternative respiration in the manuscript as “...*anaerobic respiration that utilizes alternative (to oxygen) terminal electron acceptors (hereafter alternative respiration) for substrate*”.

Comment 2. The pyrogenic carbon matrices cannot possess a potential range of 1.5 V (p.4). The potential gap between the O₂/H₂O and H⁺/H₂ redox couples is only 1.22 V (at any pH). A carbon that possesses a potential range of 1.5 V would necessarily reduce or oxidize water to H₂ or O₂, respectively.

Author response: We fully agree with the reviewer that the potential range between water splitting is 1.22 V. We had looked back at our previous publication (Sun et al., 2017) and noticed that the cited 1.5 V range actually includes H⁺ reduction. Without this, it is around 1.2 V. We have changed the value. We thank the reviewer for pointing this out.

Comment 3. Figure 1a,b: The data suggest the added char reduced CH₄ formation, probably by serving as EA. However, a quantitative electron balance was not obtained. Also, the electron donating and accepting capacities (EDC and EAC) of the chars were not measured. Based on the lower mass of CH₄ formed, one can calculate how much electron had been re-directed to respiration. How does the 147 μmol e⁻/g soil recovered (p.6) compare to the EAC of the chars (so that we would know the relative importance of chars and SOM as EA)? Additional measurements, calculations, and data analysis are needed for 400/500 °C char results.

It is clear from Figures S4 and S7 that chars prepared at 400/500 °C (as well as soil/SOM and cells of *G. sulfurreducens*) contain positive EDC. If the authors had pre-oxidized the chars with dissolved oxygen (to remove the EDC and maximize the EAC) before use, as has been done with humic materials (see, e.g., work from Sander's group at ETH), then the data would be less ambiguous and the role of chars as EA would likely be more pronounced.

Author response: In Figure 1a and b, what we found was that the pyrogenic carbon reduced CH₄ production by serving as both of an EA and ED. Pyrogenic carbon accepted electrons from microbial respiration and then donated them to SOM, based on which we identified the functional group electron transfer as a “redox-cycling electron transfer” mechanism. Regarding the EDC and EAC of pyrogenic carbon, we had already measured these values but presented them in a different way. Based on the reviewer's comment, we have now re-analyzed the results and modified the interpretation to highlight the EDC and EAC of

pyrogenic carbon in both revised manuscript and supporting information. We highly appreciate the reviewer for inspiring us to revisit our results and provide extra information. Briefly, we measured the EDC of pyrogenic carbon in the abiotic control of microcosm pure-culture incubation experiments. Based on the current data shown in Figure S7a and the calculation presented in eq. S6 and S9, Method S6, we determined the EDC of pyrogenic carbon at $0.125 \pm 0.02 \text{ mmol e}^- \text{ g}^{-1}$ pyrogenic carbon. Afterwards, pyrogenic carbon was microbially reduced by incubating it with *Geobacter sulfurreducens*. Based on the current data shown in Figure S7c and the calculation method presented in eq. S6 and S7, Method S6, we obtained the EAC of pyrogenic carbon at $0.293 \pm 0.025 \text{ mmol e}^- \text{ g}^{-1}$ pyrogenic carbon. Sum of EDC and EAC yielded an electron exchange capacity (EEC) at $0.418 \text{ mmol e}^- \text{ g}^{-1}$ pyrogenic carbon. We have now added these new calculation and interpretation in the manuscript and also presented them below to facilitate the reviewing process.

“... j_{total} in eq. S7 indicates the total oxidation current (**Figure S7c**), $j_{inoculation}$ (**Figure S7b**) and j_{HQ} (**Figure S7a**) in eq. S7 are the oxidation current induced by the background reduction. The background reduction derived from: (1) the immediate reduction of ferricyanide by the biofilm electrons after the inoculation of *G. sulfurreducens* (i.e., eq. S8); and (2) the abiotic reduction of ferricyanide by the inherent hydroquinone groups in pyrogenic carbon (i.e., eq. S9). The $j_{inoculation}$ was assessed by inoculating microbes into the growth medium that contained sand but no pyrogenic carbon and other electron acceptors. $[Ferro]_{inoculation}$ indicates the concentration of ferrocyanide resulted from the ferricyanide reduction by biofilm electrons. The j_{HQ} was determined in the growth medium with the addition of pyrogenic carbon but without microbe inoculation. $[Ferro]_{HQ}$ indicates the concentration of ferrocyanide resulted from the ferricyanide reduction by hydroquinone groups. By substituting the $[Ferro]_{redox-cycling}$ term with $[Ferro]_{HQ}$ in eq. S6, we determined the electron donation capacity (EDC) of pyrogenic carbon (produced at 500°C) at $0.125 \pm 0.02 \text{ mmol e}^- \text{ g}^{-1}$ pyrogenic carbon. By subtracting $j_{inoculation}$ and j_{HQ} from j_{total} , we obtained the oxidation current and more importantly the concentration of $[Ferro]_{redox-cycling}$. By introducing $[Ferro]_{redox-cycling}$ into eq. S6, we estimated the number ($0.293 \pm 0.025 \text{ mmol e}^- \text{ g}^{-1}$ pyrogenic carbon) of transferred electrons during the redox-cycling electron transfer of pyrogenic carbon in supporting the growth of *G. sulfurreducens*. In addition, due to the subtraction of background electrons that were donated from biofilm and hydroquinone groups, this number of transferred electrons during redox-cycling electron transfer also reflected the bioavailable electron accepting capacity (EAC) of pyrogenic carbon (produced at 500°C). The determined EDC and EAC values are in agreement with the previously reported results (Klöpffel et al., 2014)”.

$$Q = \frac{nV[Ferro]_{redox-}}{m} \quad \text{eq. S6}$$

$$[Ferro]_{redox-cycling} = \frac{j_{total} - j_{inoculation} - j_{HQ}}{0.62nFD_{Ferro}^{2/3}v^{-1/6}\omega^{1/2}} \quad \text{eq. S7}$$

$$j_{inoculation} = 0.62nFD_{Ferro}^{2/3}v^{-1/6}\omega^{1/2} [Ferro]_{inoculation} \quad \text{eq. S8}$$

$$j_{\text{HQ}} = 0.62nFD_{\text{Ferro}}^{2/3}v^{-} [\text{Ferro}]_{\text{HQ}} \quad \text{eq. S9}$$

Even though we used a different method from Sander's group to evaluate the EDC, EAC, and EEC of pyrogenic carbon, the determined values are in agreement with the results reported by Sander's group (Klüpfel et al., 2014). The reason we used the microbially reduced pyrogenic carbon to determine its EAC was to achieve a more precise access to the bioavailable EAC rather than the total EAC. We performed the electron balance analysis by comparing the number of electrons recovered from soil (i.e., the 147 $\mu\text{mol e}^-/\text{g}$ soil carbon as the reviewer noted) and the EAC of pyrogenic carbon. We found that the total number of recovered electrons was 8 times higher than the EAC, which highlighted the importance of pyrogenic carbon in facilitating microbial respiration through cycled EA and ED processes. We have clarified this importance in the manuscript as:

"...More importantly, we found that the number of accumulated electrons ($74 \pm 22 \mu\text{mol e}^-$) in the peat-soil organic matter was 8-fold ($N=3$, $P < 0.01$) higher than the electron accepting capacity ($9.1 \pm 0.7 \mu\text{mol e}^-$, SI Method S6) of pyrogenic carbon quinone groups. This excessive electron accepting suggested that after reaching its maximum capacity, pyrogenic carbon could be regenerated by donating (in the form of hydroquinone) the accepted electrons to soil organic matter..."

Comment 4. Figure 1c,d, and Figures S6,8,9,10,11: It is not clear how the "capacitive" electron transfer mechanism works, how the "capacitive" and "conductive" mechanisms differ, and whether either mechanism is relevant in actual peatland soils. The three questions are discussed below.

Comment 4.1. First, by "capacitive", do the authors mean accumulation of opposite charges at spatially separate locations of a single piece of char? I do not see how this would happen, either in reactors or in field soils. A piece of reduced char in a soil slurry or reactor would be at redox equilibrium with the surrounding solution and SOM. Since there is only one redox potential dictated by the local redox equilibrium, there would be no capacitance across a solid carbon matrix (e.g., electron exchange with aqueous SOM would remove any capacitive potential). Did the authors measure a non-zero potential across the working electrode (i.e., solid char), for example, in Figure S3b? If they did, which would support their capacitive mechanism hypothesis, please show the data.

If by "capacitive" the authors simply suggest accumulation of excess electrons in chars that contain crystalline/graphitic regions with fermi levels to accommodate electrons of different potentials, then a separate measurement should have been made to demonstrate the existence of such regions and to determine the electron holding capacity of 800 oC char. In this case, though, the char is not acting as a capacitor, and the term "capacitive" should not be used.

In any case, please clarify what is meant by "capacitive electron transfer" and how it would work exactly, and provide supporting data if available.

Author response: By capacitive, as the reviewer explained in the second paragraph of Comment 4.1, we meant the accumulation of excessive electrons in the crystalline graphitic regions of pyrogenic carbon. We had demonstrated the existence of such regions in one of our previous studies (Sun et al., 2018). Briefly, we used X-ray diffraction and scanning transmission electron microscopy coupled with electron energy loss spectroscopy to examine both of the bulk and thin sheets of pyrogenic carbon. We successfully captured the crystalline graphitic regions in pyrogenic carbon and demonstrated their structural transition, at molecular level, as a function of pyrolysis temperatures. We resolved that the crystalline/graphitic regions become more laterally expanded and connected with the increase of pyrolysis temperatures (from 600 to 800°C), which explained the increased capacitance (0.013-26 mF cm⁻² pyrogenic carbon) that was determined by cyclic voltammetric measurement (Sun et al., 2017). “Capacitive” is widely used to describe the storage phenomenon of microbial electrons by carbonaceous materials (Deeke et al., 2012; Feng et al., 2014; Houghton et al., 2016; Wang et al., 2016). We adopted this term to help a broad audience to understand. We greatly thank the reviewer for encouraging us to provide additional structural information to support our finding. We have now added this information in the manuscript. The newly added text has also been given below.

“During capacitive electron transfer, the carbon matrices stimulated alternative respiration by storing extracellular electrons in the delocalized π -electron system in the crystalline graphitic structures. Existence of crystalline graphitic structures in the carbon matrices has been demonstrated by previous studies using X-ray and electron based spectroscopic techniques (Keiluweit et al., 2010; Sun et al., 2018). Further abiotic cyclic voltammetric analysis confirmed that these structures possess a range of capacitance from 0.013 to 26 mF cm⁻² pyrogenic carbon with the increase of pyrolysis temperatures from 600 to 800°C (Sun et al., 2017)...”.

Comment 4.2. Second, it is unclear how the "capacitive" and "conductive" electron transfer mechanisms are different. Experimentally, an oxidizing (+0.5V) potential was applied to char in both systems, but only for 30 min twice a day for the former and continuously for the latter. As one would anticipate, a positive potential would favor/select for exo-electrogenic microbes, such as *G. sulfurreducens*, promoting oxidation of acetate and other substrates to CO₂, and minimizing fermentation and methanogenesis. In addition, the higher the potential, the longer it was applied, and the more conductive the char was, the more substrate oxidation and cell growth would occur. These are consistent with Figure 1c and 1d, and Figures S8, S9, S10, S11, S12 and S13.

Thus, by simply applying the same potential for different durations, it is not clear whether the authors were testing two different mechanisms. If the capacitive and conductive mechanisms really are different (in theory, not just operationally), please explain how exactly.

Author response: Both capacitive and conductive electron transfers are the electron transfer mechanisms of pyrogenic carbon matrices, especially for the pyrogenic carbon that are produced at high temperature range. Capacitive electron transfer refers to an intermittent electron transfer process that is constituted by a series of electron storage and release cycles. The capacity of electron storage and release relies on the capacitance of pyrogenic

carbon matrices. Conductive electron transfer, on the other hand, refers to a continuous electron transfer process that derives from the conductivity of pyrogenic carbon matrices. Even though we applied the same potential to test capacitive electron transfer and conductive electron transfer, the function of applied potentials and the meaning of resulting current signals were theoretically (not only operationally) different.

In capacitive electron transfer, the potential was applied only temporarily and designed to discharge pyrogenic carbon matrices. Based on the generated discharging current, we were able to quantify the number of electrons that were capacitively accumulated in pyrogenic carbon matrices. Occurrence of electron accumulation was a result of the faster electron storage rate from microbes to pyrogenic carbon matrices than electron release rate from pyrogenic carbon matrices to SOM (please also refer the Response to the Comment 4.3). Before each discharging period, no potential was applied on pyrogenic carbon, therefore the initial electron accumulation was spontaneous and relied on the activity of exo-electrogenic microbes and the capacitance of pyrogenic carbon matrices. We agree with the reviewer that an applied potential can promote the growth of exo-electrogenic microbes. For this reason, we monitored the current signals but did not observe any current increase (the solid red line in Figure 1d) before the start of capacitive electron transfer test. This indicated that the activity of exo-electrogenic microbes was not being considerably stimulated by the application of potential and the following growth of microbes was the result of capacitive electron transfer.

In contrast to capacitive electron transfer, constant potentials were applied on pyrogenic carbon during the test of conductive electron transfer. Application of constant potentials was designed to accelerate the electron release step from pyrogenic carbon matrices to terminal electron acceptors and allowed electrons to be continuously transferred through pyrogenic carbon. By this way, we were able to overcome the electron accumulation as observed in capacitive electron transfer and target on characterizing conductive electron transfer. In addition, the constantly applied potentials imposed similar effects to the reduction potentials of naturally occurring electron acceptors in accelerating electron release step from pyrogenic carbon matrices. Therefore, by applying a range of low to high potentials on pyrogenic carbon, we successfully predicted the dependency of conductive electron transfer on the type and availability of environmental electron acceptors. This dependency feature is substantially different from capacitive electron transfer, which spontaneously accumulates extracellular electrons and supports microbial respiration.

We have inserted these additional explanations into relevant sections in the supporting information Method S4 and S7, respectively. The newly added texts are also given below.

“Capacitive electron transfer refers to an intermittent electron transfer process that is constituted by a series of electron storage and release cycles. The capacity of electron storage and release relies on the capacitance of pyrogenic carbon matrices. Due to the faster electron storage rate from microbes to the carbon matrices than electron release rate from the carbon matrices to SOM (Figure S5), extracellular electrons started to accumulate in the carbon matrices. We periodically discharged the accumulated electrons

*in the carbon matrices by applying a +0.5 V (vs. SHE) electrical potential on the pyrogenic carbon WE (depicted by the red dash line in **Figure S3b**)... ”.*

“...By integrating the current as a function of incubation time, we quantified the number of accumulated electrons in the carbon matrices during capacitive electron transfer. Before each discharging period, no potential was applied on pyrogenic carbon, therefore the initial electron accumulation was spontaneous and relied on the activity of extracellular electron transfer microbes and the capacitance of pyrogenic carbon matrices... ”.

*“...We also monitored the current signals of the first 3 days of incubation but did not observe any current increase as a result of extracellular electron transfer (the solid red line in **Figure 1d** in the main text). This indicated that the activity of extracellular electron transfer microbes was not being considerably stimulated by the application of potential and the following growth of microbes was the result of capacitive electron transfer supported microbial respiration (the red dots in **Figure 1d** in the main text)... ”.*

*“Conductive electron transfer refers to a continuous electron transfer process that derives from the conductivity of pyrogenic carbon matrices. For conductive electron transfer, we applied a constant +0.5 V (vs. SHE) electrical potential on the pyrogenic carbon WE to provide a sufficient driving force and investigate the conductive electron transfer under an accelerated electron accepting condition (depicted by the red solid line in **Figure S3b**). By eliminating the electron snorkeling limit induced by the terminal electron accepting step, we were able to overcome the electron accumulation as observed in capacitive electron transfer and target on resolving the controlling effect of only conductive electron transfer on the overall electron snorkeling process... ”.*

*“For conductive electron transfer, a range of constant potentials (+0.2 to +0.5 V vs. SHE) were applied (depicted by the red solid line in **Figure S3d**) on pyrogenic carbon to investigate the limiting effect of terminal electron accepting step on electron snorkeling of conductive electron transfer. The constantly applied potentials imposed similar effects to the reduction potentials of naturally occurring electron acceptors in accelerating electron accepting step from the carbon matrices. Therefore, by applying a range of low to high potentials on pyrogenic carbon, we successfully predicted the dependency of conductive electron transfer on the type and availability of environmental electron acceptors... ”.*

Comment 4.3. Third, applying an oxidizing potential would not only support substrate oxidation and growth of *G. sulfurreducens* and other electrode-respiring bacteria, but also (abiotically) oxidize any electrode-active reductants. SOM happens to be electrode-active and would adsorb favorably to graphitic carbons, such as high temperature chars. How do we know the applied potential did not periodically or continuously re-oxidize hydroquinones and catechols (which would contribute to the observed currents) in SOM sorbed to char? This would re-generate quinones that could then serve as EA to support further respiration. It is not clear to what extent this occurred in electrochemical reactors. Hence, the "capacitive" and "conductive" mechanisms may be, at least in part, the redox cycling mechanism in disguise.

As noted above, both SOM and *G. sulfurreducens* have positive EDC (Figures S4 and S7). The applied potential would pull electron out of attached *G. sulfurreducens* cells and SOM in contact with the char, in addition to acetate (through microbial metabolism). Therefore, the observed currents most likely exaggerate the importance of the conduction mechanism. Note that (quinones in) SOM is the dominant and ultimate EA in peatland soils. Char conductivity itself cannot support respiration without SOM as EA. In the absence of an imposed potential, I don't see how char conductivity would play a significant role in situ. (Unless quinones in sorbed SOM can be respired directly through conduction, which this study does not show.)

Author response: We fully agree with the reviewer that SOM is electrode active. However, previous studies have shown that the activity of SOM at the electrode is too low to generate featured current signals in response to applied potentials (Aeschbacher et al., 2011). This is also why researchers have strived to develop new electrochemical methods that involved the usage of highly redox-active mediators to trigger and capture the redox behavior of SOM (a good review is (Sander et al., 2015)). In our system, we confirmed this low activity phenomenon by performing cyclic voltammetric scans on pyrogenic carbon with the presence of SOM (Figure S5). Results showed that no featured current peaks appeared either in oxidative or in reductive scan, indicating that the obtained current signals in bioelectrochemical soil and pure-culture incubations resulted directly from the capacitive and conductive electron transfer mechanisms instead of SOM redox mediating. Also, owing to the low redox activity between pyrogenic carbon and SOM, we were able to observe the capacitive electron transfer as we explained in the Response to the Comment 4.2. We have added this explanation in Method S4 and also given the new text below.

“By integrating the current as a function of incubation time, we were able to quantify the number of accumulated electrons in soil organic matter through conductive electron transfer. Even though soil organic matter is electrode active, its activity at the electrode is too low to generate featured current signals in response to applied potentials (Aeschbacher et al., 2011). In our system, we confirmed this low activity phenomenon by performing cyclic voltammetric scans on pyrogenic carbon with the presence of soil organic matter (Figure S5). Results showed that no featured current peaks appeared either in oxidative or in reductive scan, indicating that the obtained current signals in bioelectrochemical peat-soil incubations were directly a result from the capacitive and conductive electron transfers instead of soil organic matter mediated redox reactions...”

We agree with the reviewer that pyrogenic carbon conductivity itself cannot support respiration without EA. Therefore, we applied a range of potentials on pyrogenic carbon matrices to accelerate the electron accepting step and investigate the efficiency of conductive electron transfer under unlimited electron accepting condition. This has important implications to predict the function of conductive electron transfer in the presence of other electron acceptors (such as Fe minerals and nitrate species) which possess higher reduction potentials and can rapidly accept electrons from pyrogenic carbon through its conductivity. We have added this explanation in Method S7 and also given the new text below.

“Therefore, by applying a range of low to high potentials on pyrogenic carbon, we successfully predicted the dependency of conductive electron transfer on the type and availability of environmental electron acceptors. This prediction has important implications in interpreting the function of conductive electron transfer in the presence of electron acceptors which possess high reduction potentials (such as Fe minerals and nitrate species) and can rapidly accept electrons from pyrogenic carbon through its conductivity...”

Comment 5. Figure 2: What prompted the use of ^{13}C ? All three proposed mechanisms would not produce $^{13}\text{CO}_2$ or $^{13}\text{CH}_4$, so it is unclear why ^{13}C -enriched chars would help. How is biodegradability of chars relevant to any of the hypotheses? Please explain the rationale. It makes sense that the lower temperature chars were more labile than 800 °C char; however, biodegradation of 400/500 °C chars would likely yield both $^{13}\text{CO}_2$ and $^{13}\text{CH}_4$ (because the redox state of carbon would change from ~zero to +4 and -4) rather than favoring CO_2 over CH_4 .

Author response: We agree with the reviewer that biodegradation of pyrogenic carbon would yield emissions of both gases. This is also what our results had shown. However, previous studies have also shown that metabolism of pyrogenic carbon could modify the composition of soil microbial community (Khodadad et al., 2011), which potentially affects methanogenesis (Feng et al., 2012). Therefore, to further clarify the effect of biodegradation of pyrogenic carbon on methanogenesis activity, we performed isotopic experiments to investigate if and to what extent biodegradation of the pyrogenic carbon could contribute to overall gas emission. We found that only less than 2% of the total gas emission was derived from pyrogenic carbon, which suggested a minor effect of pyrogenic carbon on modifying microbial community and functionality. We have now added this rationale in the revised manuscript and supporting information.

Text revised in the manuscript: *“To distinguish pyrogenic carbon from native soil carbon and investigate the effect of biodegradation of pyrogenic carbon on gas production, the original biomass was isotopically labelled with ^{13}C , which resulted in a $\delta^{13}\text{C}$ of pyrogenic carbon at $774\pm 2.3\%$ (SI Method S5)...”*

Text revised in the supporting information Method S5: *“Previous studies have shown that metabolism of pyrogenic carbon could modify the composition of soil microbial community (Khodadad et al., 2011), which potentially affects methanogenesis (Feng et al., 2012). Therefore, to further clarify the effect of biodegradation of pyrogenic carbon on methanogenesis activity, we performed isotopic experiments to investigate if and to what extent biodegradation of the pyrogenic carbon could contribute to overall gas emission. In all peat-soil incubations, the added pyrogenic carbon was labelled with ^{13}C , which resulted in a $\delta^{13}\text{C}$ of pyrogenic carbon at $774\pm 2.3\%$...”*

Comment 6. SI p.10: Should be "Eq. S4 + Eq. S5" instead of "Eq. S3 + Eq. S4".

Author response: We have revised the context accordingly. We thank the reviewer for pointing out this mistake.

Comment 7. Method S3 (SI p.5): I would be careful with the assumption that pyrogenic carbon produced at 400-500 oC transferred electron through redox cycling involving quinone functional groups, whereas carbon produced at 800 oC had condensed carbon structure and therefore transferred electron through conductive solid carbon matrices.

I agree 800 oC char would be more conductive and 500 oC char would contain more quinones and other redox functional groups, but the O/C and H/C ratios of these carbons differed only by a factor of 2 (SI p.5). The EDC and EAC of these chars should have been measured and compared to assess the contributions of functional groups to the overall electron transfer. Conversely, it has been shown that small (3-6 nm) graphene islands can form at pyrolysis temperatures as low as 500 oC, which would enable electron transfer from, e.g., conductive pili and extrusions of exoelectrogenic cells to adsorbed soil organic matter, even though the bulk solid of 500 oC carbon was poorly conductive.

Author response: We agree with the reviewer that pyrogenic carbon produced at low temperature can also conductively transfer electrons and that pyrogenic carbon produced at high temperature also contains redox activity. However, what we emphasized here was the dominant electron transfer mechanism. We determined the dominance based on the comparison of electron transfer kinetics among different mechanisms instead of the quantity of transferred electrons. We found that the redox-cycling electron transfer rate of pyrogenic carbon produced at 500°C was 1000 times faster than the conductive electron transfer rate, therefore dominated the electron transfer process of low temperature pyrogenic carbon. However, this redox-cycling electron transfer rate was 10 times slower than the conductive electron transfer rate of pyrogenic carbon produced at 800°C, which indicated that conductive electron transfer was the kinetically more preferred to transfer electrons through high temperature pyrogenic carbon. We have added this explanation in Method S3. Newly added text is also given below.

*“...Further, for pyrogenic carbon produced at 500°C, we found that the redox-cycling electron transfer rate ($0.16 \text{ mmol } e^- \text{ g}^{-1} \text{ pyrogenic carbon day}^{-1}$, **Figure 3a** in the main text) was about 1000 times faster than the conductive electron transfer rate ($0.7 \text{ } \mu\text{mol } e^- \text{ g}^{-1} \text{ pyrogenic carbon day}^{-1}$, **Figure 3c** in the main text). No capacitive electron transfer process was detected in the pyrogenic carbon produced at 500°C (**Figure 3b** in the main text). Therefore, any electron snorkeling process occurred through the pyrogenic carbon that was produced in this low temperature range was mainly a result of the redox-cycling electron transfer of the functional groups...”*

Reference:

- Aeschbacher, M., Vergari, D., Schwarzenbach, R.P., Sander, M., 2011. Electrochemical Analysis of Proton and Electron Transfer Equilibria of the Reducible Moieties in Humic Acids. *Environmental Science & Technology*, 45(19): 8385-8394.
- Cruz Viggi, C. et al., 2015. The “Oil-Spill Snorkel”: an innovative bioelectrochemical approach to accelerate hydrocarbons biodegradation in marine sediments. *Frontiers in Microbiology*, 6(881).

- Deeke, A., Sleutels, T.H.J.A., Hamelers, H.V.M., Buisman, C.J.N., 2012. Capacitive Bioanodes Enable Renewable Energy Storage in Microbial Fuel Cells. *Environmental Science & Technology*, 46(6): 3554-3560.
- Feng, C., Lv, Z., Yang, X., Wei, C., 2014. Anode modification with capacitive materials for a microbial fuel cell: an increase in transient power or stationary power. *Physical Chemistry Chemical Physics*, 16(22): 10464-10472.
- Feng, Y., Xu, Y., Yu, Y., Xie, Z., Lin, X., 2012. Mechanisms of biochar decreasing methane emission from Chinese paddy soils. *Soil Biology and Biochemistry*, 46: 80-88.
- Flannigan, M.D., Amiro, B.D., Logan, K.A., Stocks, B.J., Wotton, B.M., 2006. Forest Fires and Climate Change in the 21ST Century. *Mitigation and Adaptation Strategies for Global Change*, 11(4): 847-859.
- Hoareau, M., Erable, B., Bergel, A., 2019. Microbial electrochemical snorkels (MESs): A budding technology for multiple applications. A mini review. *Electrochemistry Communications*, 104: 106473.
- Houghton, J. et al., 2016. Supercapacitive microbial fuel cell: Characterization and analysis for improved charge storage/delivery performance. *Bioresource Technology*, 218: 552-560.
- Keiluweit, M., Nico, P.S., Johnson, M.G., Kleber, M., 2010. Dynamic Molecular Structure of Plant Biomass-Derived Black Carbon (Biochar). *Environmental Science & Technology*, 44(4): 1247-1253.
- Khodadad, C.L.M., Zimmerman, A.R., Green, S.J., Uthandi, S., Foster, J.S., 2011. Taxa-specific changes in soil microbial community composition induced by pyrogenic carbon amendments. *Soil Biology and Biochemistry*, 43(2): 385-392.
- Klüpfel, L., Keiluweit, M., Kleber, M., Sander, M., 2014. Redox Properties of Plant Biomass-Derived Black Carbon (Biochar). *Environmental Science & Technology*, 48(10): 5601-5611.
- Mitov, M., Bardarov, I., Chorbadzhiyska, E., Kostov, K.L., Hubenova, Y., 2021. First evidence for applicability of the microbial electrochemical snorkel for metal recovery. *Electrochemistry Communications*, 122: 106889.
- Sander, M., Hofstetter, T.B., Gorski, C.A., 2015. Electrochemical Analyses of Redox-Active Iron Minerals: A Review of Nonmediated and Mediated Approaches. *Environmental Science & Technology*, 49(10): 5862-5878.
- Sun, T. et al., 2017. Rapid electron transfer by the carbon matrix in natural pyrogenic carbon. *Nature Communications*, 8: 14873.
- Sun, T. et al., 2018. Simultaneous Quantification of Electron Transfer by Carbon Matrices and Functional Groups in Pyrogenic Carbon. *Environmental Science & Technology*, 52: 8538-8547.
- Wang, Y., Wen, Q., Chen, Y., Yin, J., Duan, T., 2016. Enhanced Performance of a Microbial Fuel Cell with a Capacitive Bioanode and Removal of Cr (VI) Using the Intermittent Operation. *Applied Biochemistry and Biotechnology*, 180(7): 1372-1385.
- Yang, Y., Guo, J., Sun, G., Xu, M., 2013. Characterizing the snorkeling respiration and growth of *Shewanella decolorationis* S12. *Bioresource Technology*, 128: 472-478.

REVIEWERS' COMMENTS

Reviewer #4 (Remarks to the Author):

The authors have explained and/or corrected a number of minor issues I pointed out in my review. I am fine with their responses and changes.

A significant improvement in the revision that I see, which addresses my first major question about the role of 400-500 C pyrogenic carbon (PC), is the electron balance information provided. Assuming the authors' calculations are correct, the eight-fold greater electrons accepted by SOM supports the role of PC as an electron transfer mediator instead of merely an electron acceptor. The authors should, however, provide the mass loading of PC in the peat soil microcosm incubation (in Method S3 and Figure 1 and Figure S3 captions), in order for the calculations to make sense to readers.

My other concerns were also addressed, to lesser extents. Regarding the 2nd vs 3rd (capacitive vs conductive) electron transfer mechanisms proposed for PC, I accept the authors' explanations but remain unconvinced their experiments and data demonstrate both. Just because graphitic regimes exist in high-temperature PC that might permit charge accumulation in PC doesn't mean those regimes were the only or predominant entity being discharged, given that there were other chemical and microbial candidates that could do the same. I wish the authors had discharged their bioelectrochemical reactors for a much shorter time (instead of 30 minutes).

The justification for using ¹³C-enriched PC is also helpful. That section still reads like an add-on, though, as the ¹³C data really address a separate, albeit related, question (i.e., biodegradability and fate of PC) that was not the main focus of this study. Little would be lost if this section were omitted.

The revised manuscript is an improvement and may be accepted for publication.

Responses to reviewer's comments

Reviewer #4

The authors have explained and/or corrected a number of minor issues I pointed out in my review. I am fine with their responses and changes.

Author response: We thank the reviewer for the insightful and constructive comments that helped us to improve the manuscript. We are also glad that the reviewer appreciated our revision and addressing of raised issues.

Comment 1. A significant improvement in the revision that I see, which addresses my first major question about the role of 400-500 C pyrogenic carbon (PC), is the electron balance information provided. Assuming the authors' calculations are correct, the eight-fold greater electrons accepted by SOM supports the role of PC as an electron transfer mediator instead of merely an electron acceptor. The authors should, however, provide the mass loading of PC in the peat soil microcosm incubation (in Method S3 and Figure 1 and Figure S3 captions), in order for the calculations to make sense to readers.

Author response: We had already provided the mass loading of pyrogenic carbon in relevant figures and the Methods section in the main manuscript as: “The application rate was 0, 0.03 and 3 mg pyrogenic carbon g^{-1} soil in the microcosm peat-soil incubations and 0 and 1 mg pyrogenic carbon mL^{-1} growth medium in the microcosm pure-culture incubations. The application rate was 0 and 10 mg pyrogenic carbon g^{-1} soil and 0 and 6.7 mg pyrogenic carbon mL^{-1} growth medium for bioelectrochemical peat-soil and pure-culture incubations, respectively”.

Comment 2. My other concerns were also addressed, to lesser extents. Regarding the 2nd vs 3rd (capacitive vs conductive) electron transfer mechanisms proposed for PC, I accept the authors' explanations but remain unconvinced their experiments and data demonstrate both. Just because graphitic regimes exist in high-temperature PC that might permit charge accumulation in PC doesn't mean those regimes were the only or predominant entity being discharged, given that there were other chemical and microbial candidates that could do the same. I wish the authors had discharged their bioelectrochemical reactors for a much shorter time (instead of 30 minutes).

Author response: Comparing to the 11.5 hours of microbial charging time, a 30 minutes discharging time is already short. Also, based on the current profile, we found that it needed at least 30 minutes to complete a full discharging cycle.

Comment 3. The justification for using ^{13}C -enriched PC is also helpful. That section still reads like an add-on, though, as the ^{13}C data really address a separate, albeit related, question (i.e., biodegradability and fate of PC) that was not the main focus of this study. Little would be lost if this section were omitted.

Author response: We thank the reviewer for the appreciation of our rationale. We agree that the isotopic analysis section is relatively short. This is because we did not observe any microbial community change caused by degradation of pyrogenic carbon. This result is

very important though, as it further supported that the reduced methane production was a result of the electron transfer process of pyrogenic carbon rather than its utilization as a carbon source.

Comment 4. The revised manuscript is an improvement and may be accepted for publication.

Author response: We are very grateful for the reviewer's support.